# *HAMMER*: Hamiltonian Curiosity Augmented Large Language Model Reinforcement

## Abstract

Recent curriculum reinforcement learning for large language models (LLMs) typically rely on difficulty-based annotations for data filtering and ordering. However, such methods suffer from local optimization, where continual training on simple samples in the early steps causing the policy to lose its exploration. We propose a novel schema, namely Hamiltonian curiosity AugMented large language ModEl Reinforcement (HAMMER), that transfers diversity metrics, commonly used in dataset evaluation, into the dynamic reinforcement learning procedure, where training samples are ordered via a minimum-semantic Hamiltonian path making the initial training retrain more exploration. From a theoretical perspective of generalization bounds, diversity-driven ordering facilitates stable convergence. Empirical evaluations indicate that *HAMMER* stimulates model "curiosity" and consistently achieves a 3% to 4% average accuracy gain across diverse inference benchmark.

## 1 Introduction

Recently, Reinforcement Learning with Verifiable Rewards (RLVR) has emerged as a powerful tool for enhancing complex reasoning in large language models (LLMs), significantly boosting their reasoning capabilities (Luong et al., 2024; Zhang et al., 2024b; Lambert et al., 2025). During training, LLMs generate diverse responses to prompts and receive corresponding rewards (Guo et al., 2025; Shao et al., 2024; Team et al., 2025). By learning from outcome reward, these models develop the ability to produce more comprehensive reasoning traces (Chen et al., 2025b; DeepSeek-AI et al., 2025), leading to improved performance on downstream tasks. The success of large reasoning models (e.g., OpenAI-o1 (Jaech et al., 2024) and DeepSeek-R1 (DeepSeek-AI et al., 2025)) demonstrates that RLVR effectively expands the capabilities of LLMs.

Group Relative Policy Optimization (GRPO) proposed by Shao et al. (2024) is a key RLVR algorithm that extends Proximal Policy Optimization (PPO) proposed by Schulman et al. (2017), by sampling groups of responses to estimate group-relative advantages. Given reward $r$, group size $G$, policy ratio $\rho_t = \frac{\pi_\theta(o_t|q,o_{<t})}{\pi_{\theta_{\text{old}}}(o_t|q,o_{<t})}$ with bound $\varepsilon$, GRPO's objective function is

$$\mathcal{J}(\theta) = \mathbb{E}\left\{\frac{1}{G}\sum_{i=1}^{G}\frac{1}{|o_i|}\sum_{t=1}^{|o_i|}\min\left(\rho_{i,t}\hat{A}_{i,t}, clip\left(\rho_{i,t}, 1-\varepsilon, 1+\varepsilon\right)\hat{A}_{i,t}\right) - \beta\mathbb{D}_{\text{KL}}\right\},$$

where KL divergence to reference policy is $\mathbb{D}_{\text{KL}}$ with penalty factor $\beta$. The normalized advantage is $\hat{A}_{i,t} = \frac{r_{i,t}-\text{mean}(r_{i,t})}{\text{std}(r_{i,t})}$. The expectation $\mathbb{E}$ follows $(q,a) \sim \mathcal{X}$ and $\{o_i\}_{i=1}^{G} \sim \pi_\theta(\cdot|q)$. Subsequently, variants of GRPO, like Decoupled Clip and Dynamic sAmpling Policy Optimization (DAPO) (Yu et al., 2025), were proposed to optimize the GRPO.

Beyond optimization algorithms, some works explore *data-centric* strategies to improve efficiency. Inspired by human education, Curriculum Learning (CL) has been applied to LLM reinforcement (Bengio et al., 2009; Narvekar et al., 2020), most studies rely on difficulty-based sequencing of Chain-of-Thought (CoT) annotations (Parashar et al., 2025; Qiu et al., 2025). Such approaches typically mimic "easy-to-hard" progressions but require costly difficulty assessments, often via *pass@k* testing or advanced-model labeling (e.g., OpenAI-o1 (Jaech et al., 2024), Deepseek-R1 (DeepSeek-AI et al., 2025)) and suffer from local optimization. We consider adopting the diversity order, but diversity-based classical methods such as Coreset Selection (CS) (Koh & Liang, 2017;

Sener & Savarese, 2017; Lewis & Catlett, 1994; Zhang et al., 2024a) are all sampling methods whose reduction-oriented design leads to performance bottlenecks (Mehra et al., 2025).

## 1.1 MOTIVATION

Reinforcement learning with LLMs often exhibits high variance and unstable convergence, particularly in the early stages of training (Chen et al., 2025c). Traditional curriculum learning (Narvekar et al., 2020) typically follows an "easy-to-hard" strategy (Parashar et al., 2025; Qiu et al., 2025). However, in RLVR, such naive difficulty-based training often fails: **(1)** the model quickly exploits easy samples for consistent rewards, while harder ones incur repeated penalties. This early imbalance discourages exploration, leading the policy to overfit to easy problems early and become trapped in local optima, ultimately slowing convergence. Figure 5(b) confirms this inefficiency; **(2)** difficulty is a relative concept for different models, which is hard to annotate/compute; **(3)** difficulty often lies in discontinuous/uneven transfers (see Table 9(c)). To improve training, we propose a different perspective: *diversity can effectively guide RLVR*. Presenting semantically diverse samples early allows the model to explore the input space more thoroughly, reduce the generalization gap, and accelerate convergence, as theoretically justified in Section 4. In short, we transform diversity from a static dataset property to an active principle for curriculum design in LLM reinforcement learning.

## 1.2 OUR APPROACH AND CONTRIBUTIONS

In this paper, we present a novel and effective schema, *Hamiltonian Curiosity AugMented Large Language ModEl Reinforcement (HAMMER)*, which transfers diversity metrics from large model data evaluation into the dynamic process of reinforcement learning. The schema consists of two main components. First, it leverages the backbone LLM to obtain semantic similarity embeddings. Compared to external embedding models, this approach generates sentence representations that are more consistent with the model's internal training dynamics. Second, the embeddings are used to compute pairwise semantic similarity, and a Hamiltonian Curiosity Order is constructed to define a curriculum learning sequence. This process can be viewed as solving a Hamiltonian cycle that minimizes semantic similarity, enabling the model to greedily encounter the most diverse samples early in training. As a result, the model can achieve partial convergence on some samples early, improving the stability of reinforcement learning.

From a learning-theoretic perspective, we derive the generalization bound of *HAMMER*. Theorem 1 shows that early diverse training does not compromise the optimal policy, while Theorem 2 demonstrates that diverse subsets effectively tighten the generalization bound. Moreover, Theorem 3 establishes that optimizing *HAMMER*'s semantic diversity path is equivalent to maximizing the dataset diversity score, which captures the overall likelihood that a sample substantially differs from the rest of the dataset (see the formal definition in Equation 1). Extensive experiments validate our theoretical analysis and confirm its alignment with empirical results.

**Contributions**. In summary, this paper makes the following contributions:

- To improve the stability and sample efficiency of reinforcement learning, we introduce *HAMMER*, a novel curriculum learning schema. *HAMMER* structures the training sequence using a minimum semantic Hamiltonian path, termed the *Hamiltonian Curiosity Order*. This path stimulates exploratory behavior ("curiosity") in early training phases, leading to accelerated convergence and more stable optimization.

- We develop an efficient heuristic algorithm to compute the Hamiltonian Curiosity Order. This sample reordering approach delivers performance gains comparable to computationally expensive difficulty-based curriculum reinforcement learning, but with significantly lower overhead.

- From a theoretical perspective, we prove that *HAMMER* preserves the optimal policy while promoting convergence by tightening the generalization bound with a small set of diverse samples. Moreover, we show that the minimum semantic similarity cycle in HAMMER aligns with maximizing the dataset diversity score.

- Extensive experiments demonstrate that integrating *HAMMER* into RLVR algorithms like DAPO and GRPO consistently enhances sample efficiency and achieves accuracy gains of 3–4%.

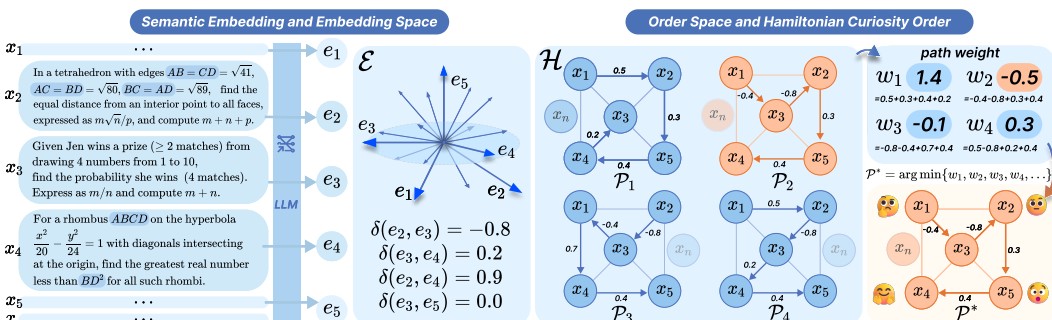

Figure 1: Overview of *HAMMER*. Given dataset $\mathcal{X} = \{x_i\}_{i=1}^{n}$, forward propagation through the backbone model yields sentence embeddings $\{e_i\}_{i=1}^{n}$, where similar ones are closer in embedding space $\mathcal{E}$ with larger similarity $\delta$ (e.g., $x_2, x_4$). Pairwise similarities form $\{\delta(e_i, e_j)\}_{n \times n}$, a complete graph. All paths of the graph consists the Order Space $\mathcal{H}$. The path $\mathcal{P}^* \in \mathcal{H}$ with minimum similarity provides the *Hamiltonian Curiosity Order*.

## 2 METHOD OVERVIEW

In this paper, we propose *Hamiltonian curiosity AugMented large language ModEl Reinforcement (HAMMER)*, a novel training schema for LLMs comprising two key components.

**Semantic Embedding** Sentence embeddings are obtained directly from the forward propagation of the backbone LLM, ensuring that the representation space reflects the model's own understanding of input text. Unlike embeddings derived from external models, this approach leverages the latent semantic structure captured by the backbone LLM itself, thereby reducing potential mismatch between training signals and the model's internal representation (BehnamGhader et al., 2024) (refer to Section 5.4 for ablation study) . Pairwise similarities between sentence embeddings define the embedding space, which can be represented as a similarity matrix $M = \{\delta(e_i, e_j)\}_{n \times n}$.

**Hamiltonian Curiosity Order** Semantic similarity matrix $M$ can be viewed as a complete graph over $n$ samples, where every edge weight corresponds to semantic proximity. All possible sample orderings in this graph form the order space, containing $n!$ distinct paths. Order space provides a rich combinatorial structure for exploring different training sequences. Within the order space, we compute the Hamiltonian cycle of minimum cumulative similarity by Algorithm 1, which we call the *Hamiltonian Curiosity Order*. This ordering intentionally prioritizes transitions across semantically dissimilar samples, thereby fostering a sense of "curiosity" in the early stages of reinforcement learning. Such curiosity-based ordering prevents premature overfitting to narrow semantic clusters, exposes the model to a broader spectrum of knowledge, and encourages more balanced exploration. As training proceeds, this induced diversity in trajectories helps smooth optimization dynamics and accelerates convergence, ultimately improving both stability and generalization of the backbone LLM under reinforcement learning. We theoretically justify this intuition in Section 4. Specifically, Theorem 1 shows that diverse subsets preserve the optimal policy during reinforcement learning. Theorem 2 further demonstrates that such subsets greedily minimize the generalization error bound. Finally, Theorem 3 establishes that the minimum semantic Hamiltonian cycle corresponds to maximizing the diversity measure $\mu_{\text{DCS}}$ (defined in Section 1).

## 3 METHODOLOGY

### 3.1 SENTENCE EMBEDDING AND SIMILARITY

Common text similarity metrics include TF-IDF (Robertson, 2004), BLEU (Papineni et al., 2002), ROUGE-L (Lin, 2004), and semantic vector similarity. While external embedding models often yield effective sentence embeddings for downstream tasks such as retrieval and classification, they may be misaligned with the backbone model's internal representations (BehnamGhader et al., 2024).

**Definition 1** (Sentence Embedding Space). *Given a dataset $\mathcal{X} = \{x_1, x_2, \ldots, x_n\}$, a sentence embedding is a mapping $f : \mathcal{X} \to \mathbb{R}^d$ (i.e., $f(x_i) = e_i$). The embedding space is $\mathcal{E} = \{f(x) : x \in \mathcal{X}\} \subset \mathbb{R}^d$, with similarity typically measured by cosine similarity: $\delta(e_i, e_j) = \frac{\langle e_i, e_j \rangle}{\|e_i\| \|e_j\|}$.*

In practice, to ensure consistency, we derive embeddings directly from the backbone LLM (BehnamGhader et al., 2024). Given a sentence $x \in \mathcal{X}$, a forward pass produces hidden states $\{h_t\}_{t=1}^{|x|}$, from which the embedding $e$ is obtained either by mean pooling over all tokens (i.e., $e = \frac{1}{|x|} \cdot \sum_{t=1}^{|x|} h_t$ ), yielding a compact vector.

**Example 1.** *As illustrated in Figure 1, each sentence in $\mathcal{X}$ is mapped into the embedding space $\mathcal{E}$ through the LLM forward pass, and similarity $\delta$ reflects semantic closeness. For instance, $x_2$ is closer in meaning to $x_4$ but more distinct from $x_3$, and their embeddings capture these relationships.*

## 3.2 HAMILTONIAN CURIOSITY DATA REORDER

**Definition 2** (Order Space). *Given a dataset $\mathcal{X} = \{x_i\}_{i=1}^n$ with embeddings $\{e_i\}_{i=1}^n$, and let the semantic similarity matrix be $M_{ij} = \frac{\langle e_i, e_j \rangle}{\|e_i\|\|e_j\|}$. Interpret $M$ as the adjacency matrix of a complete weighted graph $\mathcal{G} = (\mathcal{X}, E, \delta)$, where $E = \{(x_i, x_j) : x_i, x_j \in \mathcal{X}\}$ and $\delta(x_i, x_j) = M_{ij}$. The order space $\mathcal{H}(\mathcal{X})$ is the set of all possible sequences of the samples in $\mathcal{X}$, i.e.,*

$$\mathcal{H}(\mathcal{X}) = \Big\{ \mathcal{P} = (x_{\tau_1}, \ldots, x_{\tau_n}) : \tau \text{ is a permutation of } \{1, \ldots, n\} \Big\},$$

*where each sequence $\mathcal{P}$ corresponds to a path in $\mathcal{G}$ that visits every node exactly once.*

**Example 2.** *In Figure 1, with five samples set $\mathcal{X}_{eg} = \{x_1, x_2, x_3, x_4, x_5\}$, the order space $\mathcal{H}(\mathcal{X}_{eg})$ contains $n! = 5! = 120$ possible sequences. The figure illustrates four representative orders $\mathcal{P}_1$, $\mathcal{P}_2$, $\mathcal{P}_3$, $\mathcal{P}_4 \in \mathcal{H}(\mathcal{X}_{eg})$ . In $\mathcal{P}_1$, RL training proceeds in the order $x_1 \to x_2 \to x_5 \to x_4 \to x_3$.*

**Definition 3** (Hamiltonian Curiosity Order). *Given a path $\mathcal{P} \in \mathcal{H}(\mathcal{X})$, its cumulative similarity is defined as $w(\mathcal{P}) = \sum_{k=1}^{n-1} \delta\big(e_{\mathcal{P}_k}, e_{\mathcal{P}_{k+1}}\big)$. The Hamiltonian Curiosity Order is the Hamiltonian path $\pi^*$ that minimizes this cumulative similarity $\mathcal{P}^* = \arg\min_{\mathcal{P} \in \mathcal{H}(\mathcal{X})} w(\mathcal{P})$.*

Equivalently, $\mathcal{P}^*$ corresponds to a Hamiltonian cycle of minimum weight in $\mathcal{G}$ (Definition 2), which intentionally prioritizes traversals across semantically dissimilar samples.

**Example 3.** *In Figure 1, like Example 1, we consider samples $\mathcal{X}_{eg} = \{x_1, x_2, x_3, x_4, x_5\}$, which are embedded into $\{e_1, e_2, e_3, e_4, e_5\}$. The similarity matrix*

$$\begin{pmatrix} 1.0 & 0.5 & -0.4 & 0.7 & 0.8 \\ 0.5 & 1.0 & -0.8 & 0.9 & 0.3 \\ -0.4 & -0.8 & 1.0 & 0.2 & -0.3 \\ 0.7 & 0.9 & 0.2 & 1.0 & 0.4 \\ 0.8 & 0.3 & -0.3 & 0.4 & 1.0 \end{pmatrix}$$

*represents the weighted complete graph. Among the $5! = 120$ possible orders, the path $\mathcal{P}_2$, i.e., $x_1 \to x_3 \to x_2 \to x_5 \to x_4$ , yields the minimum cumulative similarity $w_2 = -0.4 - 0.8 + 0.3 + 0.4 = -0.5$, defining $\mathcal{P}^* = \mathcal{P}_2$. This Hamiltonian Curiosity Order ensures that the traversal moves across semantically diverse regions, thereby maximizing the diversity measure $\mu_{DCS}$ (Equation 1).*

To obtain the *Hamiltonian Curiosity Order* over the semantic similarity matrix via dynamic or integer programming is intractable for large datasets, being NP-hard (Labbé et al., 2004). Instead, we propose an *$\eta$-greedy heuristic search* ($\eta$-GHS) to efficiently approximate the minimum semantic similarity cycle, as detailed in Algorithm 1. The algorithm maintains a global best path $\mathcal{P}^*$ and its cumulative semantic similarity $w^*$ (line 3). Concretely, $\eta$-GHS performs multiple random restarts to explore diverse candidate paths ( lines 4–11), where each restart begins with a randomly selected starting node as the starting path $\mathcal{P} = \{x_0\}$ and an initialized visited set $\mathcal{V} = \{x_0\}$ (line 5). After restarting, the next node $x^*$ is chosen from the top-$\eta$ least similar unvisited nodes to encourage transitions across semantically distant samples. Then the current path and visited set are updated (lines 6–9). Upon completing a path $\mathcal{P}$, its total semantic similarity $w$ is computed (line 10) and compared with the global best $w^*$, updating it if superior (line 11). After all restarts, the algorithm returns the *Hamiltonian Curiosity Order* (line 12.) Algorithm 1 obtains a minimal semantic similarity cycle via greedy search, which is equivalent to early-stage diversity, as formalized in Theorem 3 in Section 4. The computation complexity of Algorithm 1 is $\mathcal{O}(n^2)$. Figure 4 shows a comparison of the real construction time between *HAMMER* and difficulty-based methods.

---

**Algorithm 1** Hamiltonian Cycle with Minimum Semantic Similarity

---

**Require:** dataset $\mathcal{X} = \{x_i\}_{i=1}^{n}$ with embeddings $\{e_i\}_{i=1}^{n}$, similarity matrix $M_{n \times n}$, expand factor $\eta$.
**Ensure:** reordered dataset $\mathcal{X}'$ with minimum semantic similarity.
1: $\mathcal{X}' \leftarrow$ reorder $\mathcal{X}$ by HEURISTICHAMILTON$(M, \eta)$
2: **function** HEURISTICHAMILTON$(M, \eta)$
3:     $\mathcal{P}^* \leftarrow \emptyset, w^* \leftarrow -\infty$
4:     **for** $t = 1$ to $\lfloor n/2 \rfloor$ **do**
5:        $\mathcal{P} \leftarrow$ a random $x_0 \in \mathcal{X}, \mathcal{V} \leftarrow \{x_0\}$
6:        **while** $|\mathcal{P}| < n$ **do**
7:           $x' \leftarrow$ last element of $\mathcal{P}$
8:           $x^* \leftarrow$ randomly select one of the top-$\eta$ smallest in $\{(M_{x', z}, z) : z \in \mathcal{X} \wedge z \notin \mathcal{V}\}$
9:           $\mathcal{P} \leftarrow \mathcal{P} \cup \{x^*\}, \mathcal{V} \leftarrow \mathcal{V} \cup \{x^*\}$
10:       $w \leftarrow \sum_{i=1}^{n-1} M_{\mathcal{P}_i, \mathcal{P}_{i+1}}$
11:       **if** $w > w^*$ **then** $w^* \leftarrow w, \mathcal{P}^* \leftarrow \mathcal{P}$
12:     **return** $\mathcal{P}^*$

---

### 3.3 TRAINING ON HAMILTONIAN CURIOSITY ORDERED DATASET

When trained on the Hamiltonian curiosity ordered dataset, the model greedily converges to a tighter generalization bound through subset-based training, achieving faster convergence toward the optimal policy than random shuffled dataset. We provide a theoretical justification for this phenomenon in Section 4. While such a greedy training scheme may bring little benefit to supervised learning with strong signals, it proves highly effective in the unstable setting of LLM reinforcement learning, where supervision is inherently weak. Example 4 further illustrates this idea: by greedily introducing diverse samples, *HAMMER* accelerates convergence and yields smoother training dynamics.

## 4 THEORETICAL ANALYSIS

In this chapter, we show that training on diverse subsets reduces generalization error without losing the optimal policy, forming the basis of *HAMMER*'s early diverse training, and that finding the minimal Hamiltonian cycle aligns with maximizing diversity.

### 4.1 PRELIMINARY

**Definition 4** (Optimal Policy). *Let $\mathcal{X}$ denote the sample space (e.g., state-action pairs in RL), and $\Pi$ the set of all candidate policies. For $\pi \in \Pi$, define a bounded loss function $\mathcal{L} : \Pi \times \mathcal{X} \rightarrow \mathbb{R}$. The expected risk of a policy $\pi$ is given by: $\mathcal{R}_{\mathcal{X}}(\pi) = \mathbb{E}_{x \sim \mathcal{X}}[\mathcal{L}(\pi, x)]$, while the empirical risk on a finite dataset $\mathcal{X}$ is: $\hat{\mathcal{R}}_{\mathcal{X}}(\pi) = \frac{1}{|\mathcal{X}|} \sum_{x \in \mathcal{X}} \mathcal{L}(\pi, x)$. The optimal policy $\pi^*$ is defined as the minimizer of the expected risk $\pi^* = \arg\min_{\pi \in \Pi} \mathcal{R}_{\mathcal{X}}(\pi)$.*

**Definition 5** (Induced Policy Subset). *Given the dataset $\mathcal{X}$ of size $n$, let $\mathcal{S} \subset \mathcal{X}$ and $\gamma$ be a tolerance factor. The policy subset induced by $\mathcal{S}$ is defined as $\Pi_{\mathcal{S}} = \left\{ \pi \in \Pi : \hat{\mathcal{R}}_{\mathcal{S}}(\pi) \leq \hat{\mathcal{R}}_{\mathcal{S}}^* + \gamma \right\} \subset \Pi$, where $\hat{\mathcal{R}}_{\mathcal{S}}^* = \min_{\pi \in \Pi} \hat{\mathcal{R}}_{\mathcal{S}}(\pi)$ denotes the minimal empirical risk over $\Pi$ on $\mathcal{S}$.*

**Definition 6** (Generalization Error). *Given a policy $\pi$ and the optimal policy $\pi^*$, the generalization error of $\pi$ is defined as $\Delta_\pi = |\mathcal{R}(\pi) - \mathcal{R}(\pi^*)|$.*

**Definition 7** (Diversity Metric). *Diversity metric $\mu$ is a measure from sample space to $\mathbb{R}$. The diversity $\mu(\mathcal{X})$ decreases as the sample similarity increases.*

In this work, we adopt two recent diversity metrics: *DCScore* (Zhu et al., 2025) and *n-gram* based distinct-$n$ method (Song et al., 2024b). For a dataset $\mathcal{X} = \{x_1, \ldots, x_n\}$ with embeddings $\{e_1, \ldots, e_n\}$, let $M \in \mathbb{R}^{n \times n}$ be the semantic cosine similarity matrix with $M_{ij} = \langle e_i, e_j \rangle$. The *DCScore* is defined as

$$\mu_{\text{DCS}}(\mathcal{X}) = \mathbf{tr}\left(\text{softmax}(M_{n \times n})\right) = \mathbf{tr}\left[\left(\frac{e^{M_{ij}}}{\sum_{j=1}^{n} e^{M_{ij}}}\right)_{n \times n}\right], \quad (1)$$

where softmax is applied row-wise and $\mathbf{tr}$ is the matrix trace. The $m$-gram metric measures lexical diversity by counting distinct $m$-grams across $\mathcal{X}$, let $G_m(x)$ be the multiset of $m$-grams in a sample $x$. The $m$-gram diversity is defined as

$$\mu_{\text{NGM}}(\mathcal{X}) = \frac{|\{\, g : g \in G_m(x), x \in \mathcal{X} \,\}|}{\sum_{x \in \mathcal{X}} |G_m(x)|}. \tag{2}$$

Both $\mu_{\text{DCS}}$ and $\mu_{\text{NGM}}$ may decay with increasing sample size (Zhu et al., 2025), so an adjustment is $\mu(\mathcal{X}) = |\mathcal{X}|^p \cdot \mu(\mathcal{X})$, where $p$ is a constant; following (Zhu et al., 2025), we set $p = 0.5$.

## 4.2 KEY THEOREMS

All proofs of the following theorems are detailed in Appendix B. By the VC inequality (Devroye et al., 1996) (formally defined in Lemma 1), with probability at least $1 - \delta$, for a policy class $\Pi$ with VC dimension $d$ and $n$ i.i.d. samples $\mathcal{S}$, the following inequality holds

$$\sup_{\pi \in \Pi} \left| \hat{\mathcal{R}}_{\mathcal{S}}(\pi) - \mathcal{R}(\pi) \right| \leq C \sqrt{\frac{d \log(n/d) + \log(1/\delta)}{n}}, \quad \text{where } C > 0 \text{ is some constant.} \tag{3}$$

For short, denote $\rho = C \sqrt{\frac{d \log(n/d) + \log(1/\delta)}{n}}$, where $d$ is the VC-dimension and $n$ the sample size.

**Theorem 1.** *Given a subset $\mathcal{S} \subset \mathcal{X}$ of $n$ samples, let $\pi^*$ be the optimal policy on $\mathcal{X}$. There exists some $\gamma$ (i.e., $\gamma = 2\rho$) such that $\pi^* \in \Pi_{\mathcal{S}}$.*

By Theorem 1, selecting a subset $\mathcal{S}$ from $\mathcal{X}$ that satisfies the $\gamma$-condition ensures that the optimal policy $\pi^*$ is preserved, thereby guaranteeing the optimality of the subset selection approach.

**Theorem 2.** *For a subset $\mathcal{S}$ of $n$ samples, when $\gamma = 2\rho$,*

$$\forall \pi \in \Pi_{\mathcal{S}}, \Delta_\pi \leq \mathcal{O}\left( \sqrt{\frac{d \log(n/d) + \log(1/\delta)}{n}} \right).$$

The generalization error bound $\rho \propto \mathcal{O}(\sqrt{\log n / n})$, which decreases slowly; hence, the benefit of additional samples diminishes as $n$ grows, especially in unstable LLM reinforcement learning. To this end, Theorems 1 and 2 demonstrate that, without sacrificing the optimal policy, optimizing over a subset can also effectively reduce the generalization error.

Thus, a more diverse subset $\mathcal{S}$ enables the empirical risk $\hat{\mathcal{R}}_{\mathcal{S}}$ to better approximate the true risk $\mathcal{R}$, and still enhancing generalization. We therefore partition $\mathcal{X}$ into $\mathcal{S}$ and $\mathcal{X} \setminus \mathcal{S}$, selecting $\mathcal{S}$ to maximize diversity, and adopt a two-stage training scheme. In LLM reinforcement learning, such early reduction of the generalization gap promotes convergence under high variance, since it can quickly decrease the generation error bound. This idea naturally generalizes to multi-stage: dividing $\mathcal{X}$ into $k$ subsets $\mathcal{S}_1 \subset \mathcal{X}, \mathcal{S}_2 \subset \mathcal{X}/\mathcal{S}_1, \dots, \mathcal{S}_k \subset \mathcal{X}/\cup_{i=1}^{k-1} \mathcal{S}_i$, each maximizing diversity $\mu(\mathcal{S}_i)$, and training sequentially in $k$ stages. As $k$ grows large, this process converges to our proposed *HAMMER*.

**Example 4.** *In Figure 8(a), HAMMER leverages diverse subsets (e.g., $\mathcal{S} \subset \mathcal{X}$) to rapidly reduce generalization error, while ensuring that the candidate policy set $\Pi_{\mathcal{S}}$ still retains the optimal policy $\pi^*$. Unlike training on the full dataset $\mathcal{X}$, where the generalization risk closely follows the true risk surface (green curve), training on a subset $\mathcal{S}$ yields a sparser trajectory: the subset risk does not exactly match the original risk, nor does it directly converge to the global minimum. By greedily introducing diverse samples, HAMMER accelerates convergence in LLM reinforcement learning and leads to smoother training dynamics.*

Example 4 shows how *HAMMER*, via Theorem 2 and early diversity training, quickly tightens the generalization bound and stabilizes RL. Diverse samples align empirical with true risk and reduce generalization error, enabling more efficient optimization.

**Theorem 3.** *Given a dataset $\mathcal{X} = \{x_i\}_{i=1}^n$ with embeddings $\{e_i\}_{i=1}^n$, let $\mathcal{S} \subset \mathcal{X}$ and $|\mathcal{S}| = m$, $M(\mathcal{S}) \in \mathbb{R}^{m \times m}$ be the semantic cosine similarity matrix of $\mathcal{S}$ with $M_{ij}(\mathcal{S}) = \delta(e_i, e_j)$, then we have*

$$\max_{\mathcal{S} \subset \mathcal{X}, |\mathcal{S}| = m} \mu_{DCS}(\mathcal{S}) \iff \min_{\mathcal{S} \subset \mathcal{X}, |\mathcal{S}| = m} \sum_{i=1}^m \sum_{j=1}^m \mathbb{I}(i \neq j) \cdot M_{ij}(\mathcal{S}).$$

**Example 5.** *Let $\mathcal{X}_{eg} = \{x_1, \dots, x_5\}$ and $M(\mathcal{X}_{eg})$ be the semantic-similarity matrix given in Example 3. We compare Hamiltonian Curiosity Order $\mathcal{P}^*$ and another order $\mathcal{P}_1$ of the samples:*

$$\mathcal{P}^* = \mathcal{P}_2 = (x_1, x_3, x_2, x_5, x_4) \quad \text{and} \quad \mathcal{P}_1 = (x_1, x_2, x_5, x_4, x_3).$$

Table 1: Main results comparing shuffle ordering baseline (B) and *HAMMER* (H), with $k = 100$ for AIME 2024/2025 and AMC 2023, $k = 32$ for OlympiadBench, and $\mathbf{Diff} = \mathbf{Avg_H} - \mathbf{Avg_B}$.

| Method | Dataset | pass@1 | | pass@10 | | pass@k | | cons@k | | Avg. | | Diff. |
|--------|---------|--------|--------|---------|--------|--------|--------|--------|--------|--------|--------|-------|
| | | B | H | B | H | B | H | B | H | B | H | |
| *Qwen3-1.7B* | | | | | | | | | | | | |
| DAPO | AIME 2024 | 36.3 | 39.3 | 61.0 | 64.7 | 69.0 | 74.3 | 43.3 | 43.3 | 52.4 | 55.4 | +3.0 |
| | AIME 2025 | 25.3 | 31.0 | 39.7 | 48.7 | 56.3 | 59.7 | 30.0 | 30.0 | 37.8 | 42.3 | +4.5 |
| | AMC 2023 | 64.2 | 68.9 | 83.3 | 85.1 | 90.3 | 91.3 | 74.7 | 77.1 | 78.1 | 80.6 | +2.5 |
| | OlympiadBench | 51.7 | 53.5 | 64.0 | 65.3 | 67.3 | 68.3 | 56.6 | 56.6 | 59.9 | 60.9 | +1.0 |
| GRPO | AIME 2024 | 36.3 | 40.0 | 59.3 | 63.6 | 70.0 | 73.3 | 43.3 | 43.3 | 52.4 | 55.1 | +2.7 |
| | AIME 2025 | 24.7 | 26.3 | 40.3 | 44.3 | 50.7 | 59.7 | 30.0 | 30.0 | 36.4 | 40.1 | +3.7 |
| | AMC 2023 | 63.1 | 68.5 | 83.0 | 84.7 | 88.5 | 91.2 | 74.7 | 77.1 | 77.4 | 80.4 | +3.0 |
| | OlympiadBench | 53.3 | 54.0 | 65.6 | 65.4 | 68.8 | 68.4 | 56.7 | 56.6 | 61.1 | 61.1 | 0.0 |
| *Qwen3-4B* | | | | | | | | | | | | |
| DAPO | AIME 2024 | 52.3 | 54.7 | 72.0 | 75.7 | 79.7 | 83.3 | 60.0 | 63.3 | 66.0 | 69.3 | +3.3 |
| | AIME 2025 | 39.7 | 43.7 | 51.7 | 60.7 | 63.0 | 63.3 | 46.7 | 53.3 | 50.3 | 55.3 | +5.0 |
| | AMC 2023 | 75.5 | 78.6 | 87.9 | 88.3 | 91.6 | 91.6 | 83.1 | 81.3 | 84.5 | 85.4 | +0.9 |
| | OlympiadBench | 62.4 | 63.1 | 72.8 | 74.2 | 75.5 | 76.6 | 62.5 | 64.3 | 68.3 | 69.6 | +1.3 |
| GRPO | AIME 2024 | 48.9 | 49.7 | 67.6 | 71.3 | 73.1 | 83.0 | 60.0 | 56.7 | 62.4 | 65.2 | +2.8 |
| | AIME 2025 | 40.0 | 43.7 | 54.7 | 60.3 | 60.0 | 66.3 | 53.3 | 50.0 | 52.0 | 55.8 | +3.8 |
| | AMC 2023 | 76.0 | 77.7 | 88.5 | 91.2 | 92.0 | 94.7 | 86.7 | 86.8 | 85.8 | 87.6 | +1.8 |
| | OlympiadBench | 62.5 | 63.7 | 72.5 | 74.0 | 75.4 | 76.5 | 62.7 | 64.3 | 68.3 | 69.6 | +1.3 |
| *Qwen3-8B* | | | | | | | | | | | | |
| DAPO | AIME 2024 | 58.6 | 58.3 | 78.0 | 79.0 | 81.0 | 81.7 | 60.0 | 66.6 | 69.4 | 71.4 | +2.0 |
| | AIME 2025 | 47.3 | 46.0 | 61.6 | 63.3 | 64.3 | 65.3 | 50.0 | 50.0 | 56.5 | 56.7 | +0.2 |
| | AMC 2023 | 83.9 | 81.9 | 90.0 | 91.3 | 92.2 | 92.4 | 85.5 | 85.5 | 88.5 | 88.1 | -0.4 |
| | OlympiadBench | 63.6 | 64.5 | 72.6 | 74.1 | 75.0 | 76.7 | 65.3 | 65.8 | 69.7 | 70.9 | +1.2 |
| GRPO | AIME 2024 | 59.3 | 62.3 | 79.0 | 80.0 | 81.3 | 82.3 | 60.0 | 66.6 | 70.5 | 73.4 | +2.9 |
| | AIME 2025 | 47.0 | 47.3 | 62.6 | 62.0 | 65.3 | 65.3 | 50.0 | 50.0 | 56.9 | 57.0 | +0.1 |
| | AMC 2023 | 81.9 | 81.6 | 90.6 | 91.4 | 93.0 | 92.6 | 85.5 | 85.5 | 88.3 | 88.1 | -0.2 |
| | OlympiadBench | 63.7 | 64.4 | 72.7 | 74.4 | 75.1 | 76.7 | 65.3 | 65.7 | 69.8 | 70.9 | +1.1 |
| *Deepseek-R1-Distill-Llama3-8B* | | | | | | | | | | | | |
| DAPO | AIME 2024 | 47.6 | 43.6 | 67.0 | 72.3 | 71.7 | 77.6 | 50.0 | 53.3 | 60.3 | 63.0 | +2.7 |
| | AIME 2025 | 29.3 | 30.0 | 48.3 | 50.6 | 55.6 | 57.3 | 30.0 | 33.3 | 42.7 | 44.5 | +1.8 |
| | AMC 2023 | 79.0 | 80.4 | 91.0 | 90.8 | 92.7 | 92.5 | 83.1 | 83.1 | 86.9 | 87.1 | +0.2 |
| | OlympiadBench | 55.2 | 56.9 | 68.9 | 70.5 | 72.6 | 74.3 | 58.6 | 60.2 | 64.8 | 66.4 | +1.6 |
| GRPO | AIME 2024 | 44.6 | 44.3 | 69.0 | 73.0 | 71.0 | 76.7 | 50.0 | 53.3 | 59.1 | 62.8 | +3.7 |
| | AIME 2025 | 30.7 | 33.3 | 50.0 | 50.3 | 56.0 | 57.7 | 30.0 | 33.3 | 43.2 | 45.5 | +2.3 |
| | AMC 2023 | 79.8 | 79.8 | 91.9 | 90.7 | 92.7 | 92.5 | 83.1 | 84.3 | 87.1 | 87.3 | +0.2 |
| | OlympiadBench | 55.1 | 56.6 | 69.3 | 70.7 | 72.3 | 74.2 | 58.8 | 60.0 | 64.6 | 66.2 | +1.6 |

*As defined in Equation 1, for each prefix of size $n$ we compute $\mu_{DCS} = \mathbf{tr}\big(\operatorname{softmax}(M)\big) \cdot n^p$ with $p = 1$. As shown in Table 3, we observe that for every subset ($n < 5$), the Hamiltonian curiosity order $\mathcal{P}^*$ attains a larger $\mu_{DCS}$ than $\mathcal{P}_1$, while both orders coincide on the full set ($n = 5$). Hence $\mathcal{P}_1$ maximizes sample diversity in early training stages.*

## 5 EVALUATION

### 5.1 EXPERIMENTAL SETUP

Algorithms, data and experimental details are included in supplementary materials. We evaluate *HAMMER* on four mathematical benchmarks: AIME 2024 (Li et al., 2024), AIME 2025 (Li et al., 2024), AMC 2023 (Li et al., 2024), and OlympiadBench (math ai, 2023). Models are trained on DeepScaleR (agentica org, 2023) ordered by Algorithm 1. We compare against DAPO (Yu et al., 2025) and GRPO (Shao et al., 2024) trained on randomly shuffled data. For AIME 2024/2025 and AMC 2023, we report average *pass@1, pass@10, pass@100* and *cons@100*, where *pass@k* measures solution accuracy and *cons@k* (frequency that at least one out of $k$ attempts passes verification) (frequency that the majority-answer among $k$ attempts passes verification) measures majority-vote consistency (DeepSeek-AI et al., 2025). For larger OlympiadBench, we report *pass@1*, *pass@10*, *pass@32*, and *cons@32*. See Appendix A for details.

### 5.2 MAIN EXPERIMENT

Main experiment is trained on the DeepScaleR using Qwen3-1.7B/4B/8B and Deepseek-R1-Distill-Llama3-8B as backbone models with DAPO and GRPO. The baseline adopts randomly shuffled

training data, while *HAMMER* leverages the *Hamiltonian Curiosity Order*. After convergence, we evaluate the models with *pass@1*, *pass@10*, *pass@100* and *cons@100*. As shown in Table 1, *HAMMER* achieves an average accuracy improvement of 3–4 % over the baseline. The models not only improve pass rates but also enhance answer consistency. As model size increases, the performance gains of *HAMMER* remain stable, demonstrating that *HAMMER* effectively leverages semantic similarity to optimize training efficiency without diminishing with larger models.

## 5.3 TRAINING DYNAMIC

Figure 2 and Figure 3 present the *pass@k* evaluation of Qwen3-1.7B trained on DeepScaleR across AIME 2024, AIME 2025, AMC 2023, and OlympiadBench ($k = 8$ for AIME 2024, AIME 2025, AMC 2023, and $k = 1$ for OlympiadBench). *HAMMER* consistently outperforms baselines at the same step. For GRPO on OlympiadBench, the gains are smaller but become evident in later stages. Figure 4 illustrates the training dynamics for AIME 2024 on the Qwen3-1.7B-Base, Qwen3-8B, and Deepseek-R1-Distill-Llama3-8B models using the GRPO algorithm.

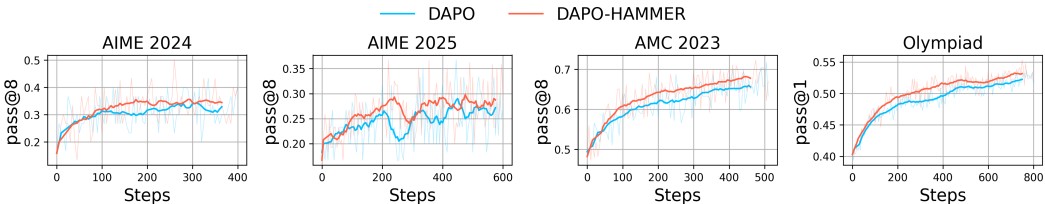

Figure 2: Validation of *pass@k* over steps on Qwen3-1.7B DAPO (8192 context).

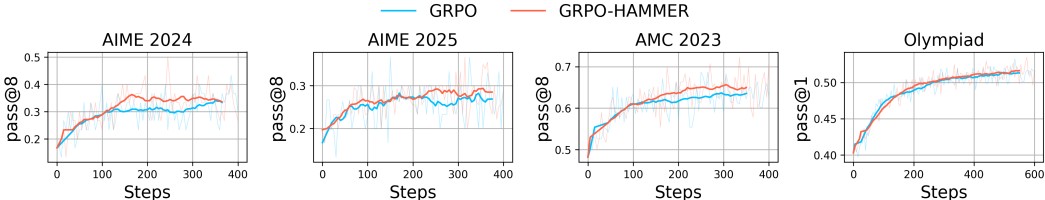

Figure 3: Validation of *pass@k* over steps on Qwen3-1.7B GRPO (8192 context).

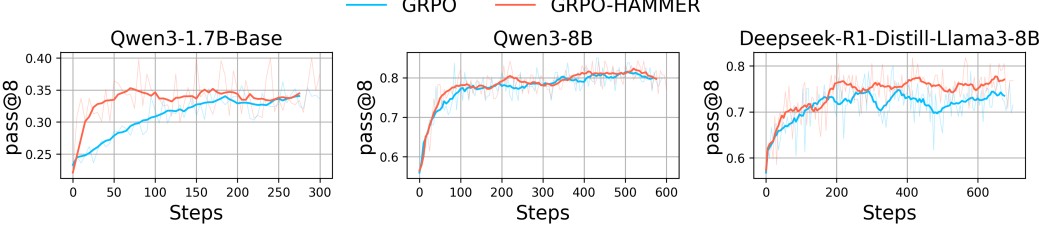

Figure 4: Validation on Qwen3-1.7B-Base/Qwen3-8B/Deepseek-R1-Distill-Llama3-8B GRPO.

## 5.4 ABLATION STUDY AND THEORETICAL VALIDATION

**Zero-shot Performance** Table 2 reports the zero-shot reasoning performance of the backbone models Qwen3-1.7B/4B/8B and Deepseek-R1-Distill-Llama3-8B on AIME 2024/2025, AMC 2023, and OlympiadBench. Combined with Table 1, while RLVR yields about a 10% improvement over the backbone models, *HAMMER* achieves a 3–4% gain solely through sample reordering.

**Maximal Semantic Sample Order** While minimal similarity ordering benefits RL, we also test maximal similarity ordering. On AIME 2024 with Qwen3-1.7B, we validate $pass@8$ using maximal semantic Hamiltonian ordering ((1)$M = -M$; (2) Algorithm 1), akin to neighbor-based training (Prashant & Easwaran, 2025). As shown in Figure 5(a), *HAMMER* remains superior.

**Difficulty-based Training** Many LLM curriculum RL approaches Parashar et al. (2025); Qiu et al. (2025) use easy-to-hard (E2H) ordering. To compare, we train DAPO on AIME 2024 with different orders. As Figure 5(b) shows, hard-to-easy (H2E) can match *HAMMER* at its peak but later unstable,

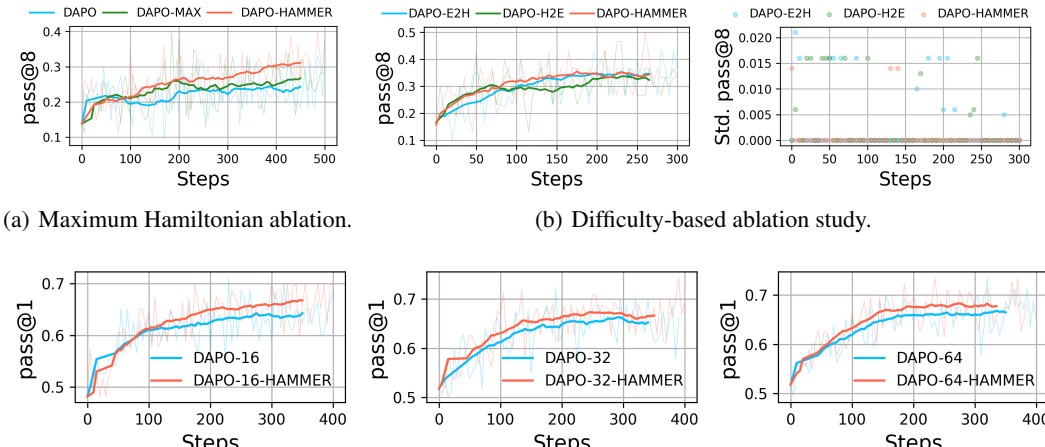

(a) Maximum Hamiltonian ablation.

(b) Difficulty-based ablation study.

(c) Validation of *pass@k* over steps on Qwen3-1.7B DAPO varying *batch size* $\in \{16, 32, 64\}$.

Figure 5: Data order and batch size ablation study, where DAPO-MAX denote *max semantic similarity* data order, DAPO-E2H and DAPO-H2E denote "easy-to-hard" and "hard-to-easy" data order.

while E2H converges slower. *HAMMER* dispenses with costly difficulty annotations, achieving robust and statistically significance with a low 8-validation standard deviation.

**Varying Batch Size** Training batch size is crucial in RLVR (Zheng et al., 2025). We vary batch size (16,32,64) with train and mini-batch sizes set equal. As shown in Figure 5(c), larger batches improve performance. *HAMMER* benefits similarly and consistently outperforms baselines by leveraging more diverse samples with bigger batch size.

**Varing Random Factor** $\eta$    $\eta$ is a random factor to generate next sample, with larger values of $\eta$ resembling random sampling. Training results show that $\eta = 3$ is the most stable configuration, outperforming $\eta = 1, 5$ (no significant difference in first 200 steps). However, $\eta = 5$ experiences a drop between steps 200–300, aligning with its closer approximation to random sampling. In contrast, $\eta = 1$ is less stable, likely due to overly strong constraints imposed by the smaller $\eta$.

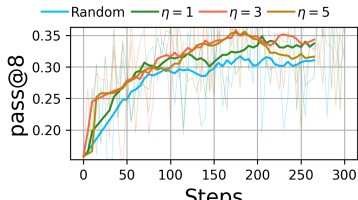

Figure 6: Ablation of $\eta \in \{1, 3, 5\}$.

**Metric Distribution** To examine how *HAMMER* reshaping affects *pass@1*, *pass@10*, *pass@100*, and *cons@100*, we evaluated three versions of the Qwen3-1.7B model across AIME 2024, AIME 2025, and AMC 2023: the base model, the DAPO-enhanced model, and DAPO further augmented with *HAMMER*. As shown in Figure 10, at the same *pass@1* level, *HAMMER* consistently improves *pass@k* for $k \geq 1$, shifting overall accuracy toward the upper-right.

Table 2: Zero-shot for AIME 2024/2025 and AMC 2023; $k = 32$ for OlympiadBench).

| Dataset | Qwen3-1.7B | | | | | Qwen3-4B | | | | |
|---|---|---|---|---|---|---|---|---|---|---|
| | pass@1 | pass@10 | pass@k | cons@k | Avg. | pass@1 | pass@10 | pass@k | cons@k | Avg. |
| AIME 2024 | 15.7 | 39.3 | 58.7 | 20.0 | 33.4 | 26.3 | 42.7 | 55.3 | 40.0 | 41.1 |
| AIME 2025 | 18.7 | 26.7 | 28.7 | 35.3 | 23.3 | 17.0 | 28.7 | 35.3 | 23.3 | 26.1 |
| AMC 2023 | 47.7 | 62.5 | 72.4 | 50.6 | 58.3 | 52.8 | 67.6 | 76.3 | 60.2 | 64.2 |
| OlympiadBench | 42.1 | 53.3 | 55.7 | 43.6 | 48.6 | 45.5 | 55.3 | 58.3 | 46.5 | 51.4 |

| Dataset | Qwen3-8B | | | | | Deepseek-R1-Distill-Llama3-8B | | | | |
|---|---|---|---|---|---|---|---|---|---|---|
| | pass@1 | pass@10 | pass@k | cons@k | Avg. | pass@1 | pass@10 | pass@k | cons@k | Avg. |
| AIME 2024 | 34.0 | 52.6 | 56.6 | 33.3 | 45.2 | 27.3 | 51.0 | 62.6 | 23.2 | 44.0 |
| AIME 2025 | 19.6 | 36.3 | 40.0 | 16.7 | 31.0 | 19.0 | 40.3 | 46.6 | 20.0 | 33.1 |
| AMC 2023 | 58.7 | 74.2 | 76.8 | 59.0 | 67.8 | 62.7 | 83.0 | 86.8 | 63.8 | 75.1 |
| OlympiadBench | 45.3 | 54.5 | 57.2 | 44.4 | 51.0 | 42.2 | 57.4 | 61.2 | 43.5 | 52.0 |

**Theory Validation** To validate Theorem 3 and explaination in Example 5, we compute $\mu_{\text{DCS}}$ and $\mu_{\text{NGM}}$ on DeepScaleR under varying subset ratios. Figure 9 show that *HAMMER* prioritizes the most diverse samples with the same subset scale;

## 5.5 EXPLORATION AND DISCUSSION

**Comparison with other CL Methods** We compare *HAMMER* with three recent CL approaches, ADARFT (Shi et al., 2025), SEC (Chen et al., 2025a), and E2H-G/C (Parashar et al., 2025), which are summarized in Section D. To ensure fairness given differing model and dataset settings, we conduct two comparisons: (1) Qwen2.5-Math-1.5B using the ADARFT settings; (2) Qwen2.5-1.5B-Instruct using SEC (Chen et al., 2025a) settings. As shown in Table 5, *HAMMER* outperforms ADARFT in first experiment, which cannot be explained by GRPO alone. In the second experiment, *HAMMER* outperforms E2H-C and is slightly below E2H-G. However, *HAMMER* avoids the need for difficulty annotations , offering a more low-cost/efficient training strategy (Table 4).

**Comparison with other Embedding Models** To assess the effect of embedding models, we train on DeepScaleR and validate on AMC 2023, using all-MiniLM-L6-v2 (BERT) and Qwen3-Embedding-4B to generate curriculum orders via Algorithm 1. Figure 7(a) shows that *HAMMER*'s backbone embedding consistently outperforms external encoders: BERT and Qwen3-Embedding-4B match HAMMER only in the first 100–200 steps but eventually converge to random ordering.

**Combination E2H with *HAMMER*** To examine the effect of combining E2H and *HAMMER*, we construct a new curriculum for DeepScaleR (E2H&HAMMER) using difficulty as the primary key and *HAMMER*'s diversity order as the secondary key, and evaluate it on AIME 2024. Figure 7(b) shows that this combined ordering underperforms both individual methods and resembles random.

**Mini-batch Selection with Diversity Constraint** Section 4 inspires that diversity subsets can also be applied locally in mini-batch selection. To examine this, we train on DeepScaleR and validate on AMC 2023, selecting 50% of each mini-batch for backward using the farthest-point strategy (Coreset Selection in Appendix D). Figure 7(c) shows that this diversity-based selection uses fewer samples yet surpasses full-batch training, supporting Theorems 1 and 2. However, as discussed in Section 1, such coreset-style selection can impose performance bottleneck; accordingly, *HAMMER*'s global ordering yields larger gains than the mini diversity constraint batch method (MCB).

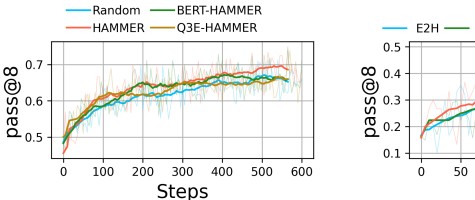 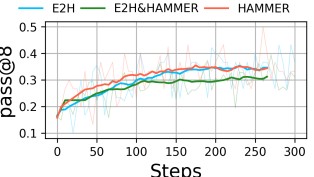 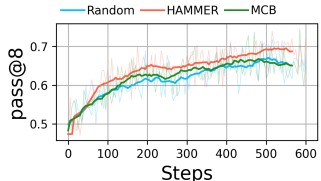

(a) Embedding model ablation.    (b) Combine E2H with *HAMMER*.    (c) Diversity constraint mini-batch.

Figure 7: Training dynamics for additional exploration studies. Q3E denotes Qwen3-Embedding-4B. E2H&HAMMER prioritizes difficulty followed by diversity. MCB (mini-constraint batch) selects the top 50% most diverse samples using BERT sentence embeddings.

## 6 CONCLUSION

We present *HAMMER*, a novel schema that integrates semantic diversity into reinforcement for LLMs. By leveraging a minimum-semantic Hamiltonian path to define a curriculum sequence, *HAMMER* stimulates early-stage model "curiosity", accelerates convergence, and improves training stability. Theoretically, we show that *HAMMER* preserves the optimal policy while tightening generalization bounds through diverse sample, and that minimizing semantic similarity aligns with maximizing the dataset diversity measure $\mu_{\text{DCS}}$. Empirically, *HAMMER* consistently enhances sample efficiency across multiple benchmarks, yielding 3%–4% average accuracy gains, demonstrating the effectiveness and generality of diversity-driven curriculum learning in LLM reinforcement training.

## REPRODUCIBILITY STATEMENT

Algorithms, data and experimental details are included in the anonymous repository `https://anonymous.4open.science/r/HAMMER-B17F` and provided as supplementary material.

## ETHICS STATEMENT

All datasets used are publicly available with appropriate licenses. Our method is designed to improve LLM training efficiency, and should be used responsibly. We do not expect our work to produce harmful content, and encourage ethical deployment in line with the ICLR Code of Ethics.

## THE USE OF LARGE LANGUAGE MODELS (LLMS)

Although the paper proposes a method to improve the training efficiency of LLMs. LLMs were used only to aid and polish the writing. No part of the research, method, or experiments relied on LLMs. The authors take full responsibility for the paper.

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

## A   EXPERIMENTAL SETUP

**Datasets**   All datasets are detailed in Table 8 in Appendix E. We evaluate our method on four benchmark datasets for mathematical problem solving. AIME 2024 (30 problems) (Li et al., 2024), AIME 2025 (30 problems) (Li et al., 2024), AMC (83 problems) (Li et al., 2024) and OlympiadBench (675 problems) (math ai, 2023). All models are trained on the DeepScaleR (40,315 problems), which provides high-quality synthetic reasoning traces designed to enhance step-by-step mathematical reasoning. In *HAMMER*, DeepScaleR is ordered by minimal similarity using Algorithm 1. The embedding of $\mathcal{X}$ is computed by mean pooling. We set $\eta = 3$ refer to the parameter ablation study in Section 5.4 and Figure 6.

**Baselines**   For the main experiments (Table 1), we use DAPO and GRPO trained on randomly shuffled samples as baselines, while *HAMMER* differs in the sample ordering. Appendix E provides details about backbone models.

**Training**   In our experiment, we adapt Qwen3-1.7B-Base, Qwen3-1.7B/4B/8B (Yang et al., 2025) and Deepseek-R1-Distill-Llama3-8B-DAPO/GRPO DeepSeek-AI et al. (2025) as the backbone model, and train on *verl* (Sheng et al., 2024) through GRPO (Shao et al., 2024) and DAPO (Yu et al., 2025). Models for main experiment were trained with a batch size of 16 (including mini-batch size). The maximum prompt length is set to 1024 tokens, and the maximum response length is 8192 tokens. For the training hyper-parameters, learning rate is fixed at $1 \times 10^{-6}$ without warmup step. For GRPO we adopt KL regularization (coefficient $\beta = 0.001$). For DAPO, we set $\varepsilon_{\text{low}} = 0.2$ and $\varepsilon_{\text{high}} = 0.28$ and token-level policy gradient loss, and dynamically filter samples by accuracy during training. Each training step generates 16 rollouts, while validation (in dynamic experiments) uses 8 rollouts, ensuring stability (e.g., the low standard deviation shown in Figure 5(b)) . The rollout temperature is 1.2, and the validation temperature is 0.6. For the reward, if the $i$-th rollout passes verification, it is assigned a positive reward $r_i = 1$; otherwise, it receives $r_i = 0$. Additionally, we provide comparative experiments of the Qwen3-1.7B model with other RLVR algorithms (i.e., REINFORCE++) and non-RLVR algorithms (i.e., DPO) in Table 6, which validate *HAMMER*'s extensibility to different algorithms. To ensure training stability, we present multiple training results in Table 7. Although there are fluctuations during training, the overall stability and performance gains brought by *HAMMER* are significant.

**Evaluation**   To evaluate LLM performance, we set temperature to 1.2, with top-$p = 0.95$ and top-$k = 20$ with 8192 context length. For AIME 2024, AIME 2025, and AMC 2023, we sample 1, 10, and 100 responses 10 times and report average $pass@k$ ($k \in 1, 10, 100$) and $cons@100$, measuring solution accuracy and response consistency (DeepSeek-AI et al., 2025). For OlympiadBench, due to its larger size, we evaluate only $pass@1$, $pass@10$, $pass@32$, and $cons@32$, which are sufficient for reliable estimation.

## B   PROOF AND DISCUSSION OF SECTION 4

### B.1   PROOF OF SECTION 4

**Lemma 1** (Vapnik-Chervonenkis Inequality). *For a policy class $\Pi$ with VC dimension $d$ and $n$ i.i.d. samples $\mathcal{S}$, the following inequality holds*

$$\forall \varepsilon \in \mathbb{R}^+, \quad \mathbb{P}\left(\sup_{\pi \in \Pi}\left|\hat{\mathcal{R}}_{\mathcal{S}}(\pi) - \mathcal{R}(\pi)\right| \geq \varepsilon\right) \leq 2\left(\frac{en}{d}\right)^d e^{-n\varepsilon^2/2}.$$

*Setting $\mathbb{P}$ to $\delta$ and solving for $\varepsilon$ yields the generalization bound.*

$$\sup_{\pi \in \Pi}\left|\hat{\mathcal{R}}_{\mathcal{S}}(\pi) - \mathcal{R}(\pi)\right| \leq C\sqrt{\frac{d\log(n/d) + \log(1/\delta)}{n}}, \quad \text{where } C > 0 \text{ is some constant.}$$

*Proof.* The proof of Lemma 1 relies on *Hoeffding's inequality* (Devroye et al., 1996). Setting $\mathbb{P}$ to $\delta$ and solving for $\varepsilon$ yields the generalization bound (shorten $C\sqrt{\frac{d\log(n/d) + \log(1/\delta)}{n}}$ as $\rho$)   □

**Theorem 1.** *Given a subset $\mathcal{S} \subset \mathcal{X}$ of $n$ samples, let $\pi^*$ be the optimal policy on $\mathcal{X}$. There exists some $\gamma$ (i.e., $\gamma = 2\rho$) such that $\pi^* \in \Pi_{\mathcal{S}}$.*

*Proof.* Let $\gamma = 2\rho$. From Inequality 3, we have the uniform bound

$$\forall \pi \in \Pi, \left| \hat{\mathcal{R}}_{\mathcal{S}}(\pi) - \mathcal{R}(\pi) \right| \leq \rho. \tag{4}$$

Particularly, the optimal policy $\pi^*$ yields

$$\hat{\mathcal{R}}_{\mathcal{S}}(\pi^*) \leq \mathcal{R}(\pi^*) + \rho. \tag{5}$$

Moreover, applying the uniform bound to all policies and taking the minimum

$$\hat{\mathcal{R}}_{\mathcal{S}}^* = \min_{\pi \in \Pi} \hat{\mathcal{R}}_{\mathcal{S}}(\pi) \geq \min_{\pi \in \Pi}[\mathcal{R}(\pi) - \rho] = \min_{\pi \in \Pi} \mathcal{R}(\pi) - \rho = \mathcal{R}(\pi^*) - \rho. \tag{6}$$

Combining Inequality 5 and 6, we obtain

$$\hat{\mathcal{R}}_{\mathcal{S}}(\pi^*) - \hat{\mathcal{R}}_{\mathcal{S}}^* \leq (\mathcal{R}(\pi^*) + \rho) - (\mathcal{R}(\pi^*) - \rho) = 2\rho.$$

Thus, $\hat{\mathcal{R}}_{\mathcal{S}}(\pi^*) \leq \hat{\mathcal{R}}_{\mathcal{S}}^* + \gamma$, which by Definition 5 implies $\pi^* \in \Pi_{\mathcal{S}}$. $\qquad\square$

**Theorem 2.** *For a subset $\mathcal{S}$ of $n$ samples, when $\gamma = 2\rho$,*

$$\forall \pi \in \Pi_{\mathcal{S}}, \Delta_\pi \leq \mathcal{O}\left(\sqrt{\frac{d \log(n/d) + \log(1/\delta)}{n}}\right).$$

*Proof.* $\forall \pi \in \Pi_{\mathcal{S}}$, by Inequality 4 and Definition 5, we have

$$\mathcal{R}(\pi) \leq \hat{\mathcal{R}}_{\mathcal{S}}(\pi) + \rho \leq \left(\hat{\mathcal{R}}_{\mathcal{S}}^* + \gamma\right) + \rho = (\hat{\mathcal{R}}(\pi^*) + \gamma) + \rho = \hat{\mathcal{R}}(\pi^*) + 3\rho$$

Thus, we deduce $\Delta_\pi = |\mathcal{R}(\pi) - \mathcal{R}(\pi^*)| \leq 3\rho = \mathcal{O}\left(\sqrt{\frac{d \log(n/d) + \log(1/\delta)}{n}}\right)$. $\qquad\square$

**Theorem 3.** *Given a dataset $\mathcal{X} = \{x_i\}_{i=1}^n$ with embeddings $\{e_i\}_{i=1}^n$, let $\mathcal{S} \subset \mathcal{X}$ and $|\mathcal{S}| = m$, $M(\mathcal{S}) \in \mathbb{R}^{m \times m}$ be the semantic cosine similarity matrix of $\mathcal{S}$ with $M_{ij}(\mathcal{S}) = \delta(e_i, e_j)$, then we have*

$$\max_{\mathcal{S} \subset \mathcal{X}, |\mathcal{S}|=m} \mu_{DCS}(\mathcal{S}) \iff \min_{\mathcal{S} \subset \mathcal{X}, |\mathcal{S}|=m} \sum_{i=1}^m \sum_{j=1}^m \mathbb{I}(i \neq j) \cdot M_{ij}(\mathcal{S}).$$

*Proof.* First, clarify the softmax convention: let $\mathrm{softmax}$ denote the row-wise softmax applied to matrix $M_{n \times n}$, i.e.

$$\mathbb{P}_{ij}(\mathcal{S}) = \frac{e^{M_{ij}(\mathcal{S})}}{\sum_{k=1}^m e^{M_{ik}(\mathcal{S})}} \qquad \text{for } i, j = 1, \ldots, m.$$

By definition $\mu_{\mathrm{DCS}}(\mathcal{S}) = \mathbf{tr}(\mathbb{P}(\mathcal{S})) = \sum_{i=1}^m \mathbb{P}_{ii}(\mathcal{S})$. Using the fact that each row of $\mathbb{P}(\mathcal{S})$ sums to 1, we have for any fixed $\mathcal{S}$ $\mathbb{P}_{ii}(\mathcal{S}) = 1 - \sum_{j \neq i} \mathbb{P}_{ij}(\mathcal{S})$, and therefore

$$\mu_{\mathrm{DCS}}(\mathcal{S}) = \sum_{i=1}^m \mathbb{P}_{ii}(\mathcal{S}) = m - \sum_{i=1}^m \sum_{j \neq i} \mathbb{P}_{ij}(\mathcal{S}).$$

Hence maximizing $\mu_{\mathrm{DCS}}(\mathcal{S})$ is equivalent to minimizing the total off-diagonal mass of $\mathbb{P}(\mathcal{S})$:

$$\max_{\mathcal{S} \subset \mathcal{X}, |\mathcal{S}|=m} \mu_{\mathrm{DCS}}(\mathcal{S}) \iff \min_{\mathcal{S} \subset \mathcal{X}, |\mathcal{S}|=m} \sum_{i=1}^m \sum_{j \neq i} \mathbb{P}_{ij}(\mathcal{S}).$$

Next, relate the off-diagonal entries $\mathbb{P}_{ij}(\mathcal{S})$ to the original similarity values $M_{ij}(\mathcal{S})$. For each fixed row $i$, $\mathbb{P}_{ij}(\mathcal{S})$ is a strictly increasing function of $M_{ij}(\mathcal{S})$ (holding the other entries in the same row

fixed). In particular, increasing any off-diagonal similarity $M_{ij}$ (with other row-$i$ entries unchanged) strictly increases the corresponding $\mathbb{P}_{ij}$ and thus increases the row's off-diagonal mass $\sum_{j \neq i} \mathbb{P}_{ij}$. Consequently, a subset $\mathcal{S}$ that yields smaller off-diagonal similarity values $M_{ij}$ will also yield smaller total off-diagonal mass $\sum_{i \neq j} \mathbb{P}_{ij}$, and hence larger $\mu_{\text{DCS}}(\mathcal{S})$.

Combining the two steps above, we obtain the stated equivalence at the level of optimization over subsets:

$$\max_{\mathcal{S} \subset \mathcal{X}, |\mathcal{S}|=m} \mu_{\text{DCS}}(\mathcal{S}) \quad \Longleftrightarrow \quad \min_{\mathcal{S} \subset \mathcal{X}, |\mathcal{S}|=m} \sum_{i=1}^{m} \sum_{j=1}^{m} \mathbb{I}(i \neq j) \cdot M_{ij}(\mathcal{S}).$$

$\square$

## B.2 DISCUSSION OF SECTION 4

**Discussion 1** ( Role of Theorem 1). *Theorem 1, derived from the VC inequality (1), establishes that under the confidence δ, we can always find subset $\mathcal{S}$, which preserves the optimal strategy. Rather than verifying whether a given $\mathcal{S}$ satisfies $\gamma \geq 2\rho$, the theorem provides a flexibility for identifying a $\mathcal{S}$ on which training alone does not harm the optimality. When the model's data tolerance sufficient ($\gamma \geq 2\rho$), Theorem 1 holds and ensures the non-volatility of the optimal strategy during subset learning. Practically, LLMs exhibit this considerable tolerance to data scale (Li et al., 2025a). Theorem 1 thus offers a theoretical justification for this observation and serves as the basis for Theorem 2 and Theorem 3.*

**Discussion 2** ( Role of Theorem 2). *Theorem 2 shows that when the subset $\mathcal{S}$ satisfies the data tolerance $\gamma \geq 2\rho$ , the generalization risk from training on $\mathcal{S}$ can be bounded within a constant factor (i.e., $3\times$) of the true generalization risk. Together, Theorems 1 and 2 support that one can often train only on a subset $\mathcal{S}$ without losing performance.*

**Discussion 3** ( Theorem 3 and Hamiltonian Curiosity Order). *Zhu et al.; Leinster & Cobbold claim that diversity remains constant under any data permutation, highlighting that static diversity are not directly unsuitable for ordering. HAMMER novelly uses local diversity to build dynamic curriculum:*

* *While overall diversity remains fixed, local diversity within subsets can change: increasing diversity in some samples corresponds to decreasing diversity in others.*

* *The model does not see all samples at once; instead, it gradually encounters the data, meaning local diversity impacts training dynamics:*

*Two natural strategies are proposed: (1) First, train on a diverse subset $\mathcal{S}$, then train on the remaining similar samples $\mathcal{X} \setminus \mathcal{S}$ (e.g., HAMMER); (2) First, train on a similar subset, then train on the more diverse one. Figure 5(a) compares these two approaches.*

*Based on Theorem 1 and 2, we find that training on subset $\mathcal{S}$ can also significantly reduce the generalization error, while diversity improves model robustness (Zhu et al., 2025), bringing the error closer to the true error. Theorem 3 further provides an equivalence between diversity and similarity, leading to HAMMER: train on samples with minimal semantic similarity first.*

*Additionally, we greedily partition $\mathcal{X}$ into $\mathcal{S}_1 \subset \mathcal{X}$, $\mathcal{S}_2 \subset \mathcal{X} \setminus \mathcal{S}_1$, ..., $\mathcal{S}_k \subset \mathcal{X} \setminus \bigcup_{i=1}^{k-1} \mathcal{S}_i$, where each subset maximizes its diversity $\mu(\mathcal{S}_i)$. If the model is still robust to subset size (Li et al., 2025a), training on "minimal semantic similarity" subsets helps improve generalization. This process is equivalent to solving a greedy approximation of the global "minimal semantic similarity Hamiltonian circuit" (Algorithm 1), with $\eta$ balancing the trade-off between greedy selection and random shuffling.*

*Although the overall diversity of the dataset remains unchanged, Figure 8(b) shows that HAMMER, compared to random ordering, places more diverse samples in the early stages of local learning, which is consistent with Example 5 and Figure 9. Numerous experiments and the above analysis have demonstrated the effectiveness of constructing the curriculum with HAMMER.*

Table 3: Comparison of $\mu_{\text{DCS}}$ values for different prefix subsets under two orders.

| Prefix Subset | $\{\mathcal{P}_1\}$ | $\{\mathcal{P}_1, \mathcal{P}_2\}$ | $\{\mathcal{P}_i\}_{i=1}^3$ | $\{\mathcal{P}_i\}_{i=1}^4$ | $\{\mathcal{P}_i\}_{i=1}^5$ |
|---|---|---|---|---|---|
| $\mu_{\text{DCS}}$ of $\mathcal{P}^*$ | 1.00 | 3.43 | 5.59 | 6.78 | 8.35 |
| $\mu_{\text{DCS}}$ of $\mathcal{P}_1$ | 1.00 | 2.49 | 3.96 | 5.24 | 8.35 |

## C  SUPPLEMENTARY TABLES AND FIGURES

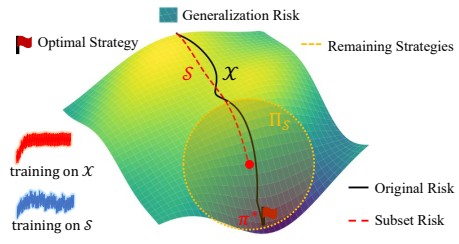

(a) Illustration of earlier diversity training.

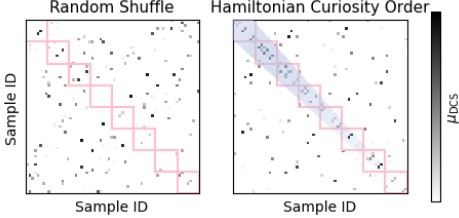

(b) Distribution of $\mu_{\text{DCS}}$; pink boxes denote training mini-batches following the diagonal sample order.

Figure 8: Illustration of Example 4 and distribution of $\mu_{\text{DCS}}$.

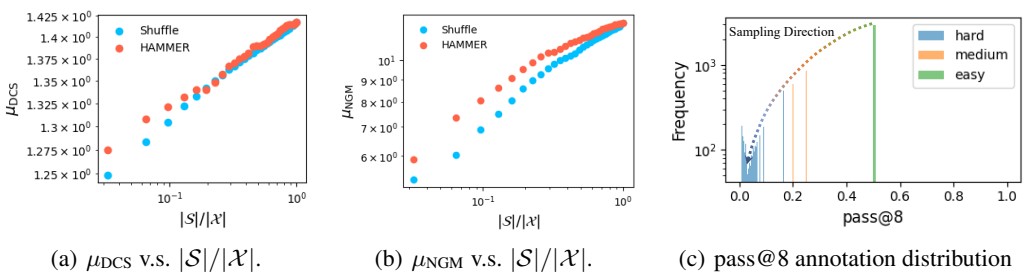

(a) $\mu_{\text{DCS}}$ v.s. $|\mathcal{S}|/|\mathcal{X}|$.    (b) $\mu_{\text{NGM}}$ v.s. $|\mathcal{S}|/|\mathcal{X}|$.    (c) pass@8 annotation distribution

Figure 9: $\mu$ (for DeepScaleR) varying different subset ratios and pass@8 annotation distribution.

Table 4: Time comparison for training dataset ordering (minutes) : S1 (Embedding), S2 (Similarity Matrix), and S3 (Hamiltonian Ordering).

| Method | Model | S1 | S2 | S3 | Total |
|---|---|---|---|---|---|
| HAMMER | Qwen3-1.7B | 16.52 | 94.95 | 13.02 | 124.48 ↓ |
| | Qwen3-4B | 27.72 | 95.32 | 13.30 | 136.33 ↓ |
| | Qwen3-8B | 42.95 | 95.68 | 13.38 | 152.02 ↓ |
| | Deepseek-R1-Distill-Llama3-8B | 46.96 | 96.02 | 12.90 | 155.88 ↓ |
| E2H (8 rollout → sort) | Qwen3-1.7B | | | | 195.78 |
| | Qwen3-4B | | | | 334.48 |
| | Qwen3-8B | | | | 538.51 |
| | Deepseek-R1-Distill-Llama3-8B | | | | 542.32 |

## D  RELATED WORKS

**RL for LLM Reasoning** Reinforcement Learning (RL) plays a critical role in LLM post-training. Traditional methods such as Reinforcement Learning from Human Feedback (RLHF) train a reward model to compare response preferences and optimize policy via algorithms like Proximal Policy Optimization (PPO) (Schulman et al., 2017). To simplify, Direct Preference Optimization (DPO) (Rafailov et al., 2023) was proposed, which directly optimizes policy using pairwise preference

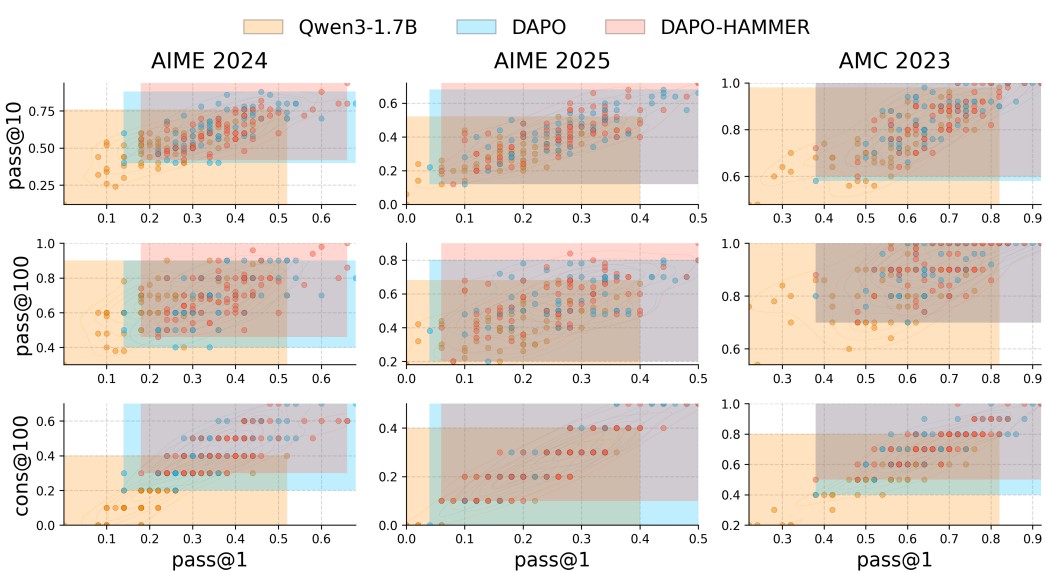

Figure 10: Distribution of metrics.

Table 5: Comparison against other curriculum learning methods, with 8-shot (pass@8) on AIME 2024 & AMC 2023 and 1-shot (pass@1) on others.

| Data Split | Algorithm | Comparison with ADARFT | | | | | ▷ Qwen2.5-Math-1.5B |
|---|---|---|---|---|---|---|---|
| | | GSM8K | AIME 2024 | AMC 2023 | OlympiadBench | Avg. | Source |
| full | Zero-shot | 43.5 | 5.3 | 22.8 | 18.2 | 22.5 | Ours |
| skew-difficult | PPO | 69.7 | 9.2 | 47.5 | 20.7 | 36.7 | Shi et al. |
| skew-difficult | PPO (w/ Filter) | 71.7 | 9.2 | 45.0 | 20.1 | 36.5 | Shi et al. |
| skew-difficult | ADARFT (PPO) | 74.0 | 12.1 | 55.0 | 20.4 | 40.4 | Shi et al. |
| uniform | PPO | 72.0 | 6.7 | 42.5 | 21.1 | 35.6 | Shi et al. |
| uniform | PPO (w/ Filter) | 72.6 | 10.0 | 45.0 | 20.2 | 37.0 | Shi et al. |
| uniform | ADARFT (PPO) | 74.5 | 12.1 | 57.5 | 22.0 | 41.5 | Shi et al. |
| skew-easy | PPO | 72.7 | 12.5 | 45.0 | 19.2 | 37.4 | Shi et al. |
| skew-easy | PPO (w/ Filter) | 74.8 | 10.0 | 45.0 | 20.4 | 37.5 | Shi et al. |
| skew-easy | ADARFT (PPO) | 74.5 | 9.2 | 55.0 | 19.9 | 39.7 | Shi et al. |
| full | GRPO (Random) | 72.4 | 12.2 | 55.1 | 20.5 | 40.1 | Ours |
| full | GRPO (*HAMMER*) | 77.8 | 15.3 | 58.2 | 23.0 | 43.6 | Ours |

| Data Split | Algorithm | Comparison with SEC/E2H-G(C) (GSM8K) | | | | | ▷ Qwen2.5-1.5B-Instruct |
|---|---|---|---|---|---|---|---|
| | | Trivial | Easy | Med | Hard | Overall. | Source |
| full | Zero-shot | NA | NA | NA | NA | 73.2 | Qwen et al. |
| 4 levels | SEC | 98.1 | 95.3 | 87.0 | 50.3 | 77.8 | Parashar et al. |
| 4 levels | E2H-G | 97.6 | 94.7 | 89.0 | 51.8 | 78.7 | Parashar et al. |
| 4 levels | E2H-C | 98.0 | 95.3 | 83.9 | 46.6 | 75.7 | Parashar et al. |
| full | GRPO (Random) | NA | NA | NA | NA | 75.2 | Ours |
| full | GRPO (*HAMMER*) | NA | NA | NA | NA | 78.2 | Ours |

data without reward model. A key development is Reinforcement Learning with Verifiable Rewards (RLVR), which offer feedback based on outcome correctness or verifiable reward, significantly enhancing LLMs' reasoning in mathematics and programming (Li et al., 2025b). OpenAI's o1 (Jaech et al., 2024) advanced reasoning, while DeepSeek-R1 (DeepSeek-AI et al., 2025)introduced zero-RL by eliciting model's slow-thinking capability. These advances spurred Large Reasoning Models (LRMs) like Kimi 1.5 (Team et al., 2025) and QwQ (Team, 2025). A key RLVR algorithm, Group Relative Policy Optimization (GRPO), extends PPO by sampling multiple responses to compute group-relative advantage, yielding major gains (Shao et al., 2024). It inspired variants like DAPO (Yu et al., 2025) , VAPO (Yan et al., 2025) and GSPO (Zheng et al., 2025).

**Curriculum RL** Curriculum Learning (CL) takes inspiration from human education, structuring learning from simple to complex concepts (Narvekar et al., 2020). Recent work has applied

Table 6: Comparison of REINFORCE++ and DPO under Random vs. HAMMER sampling, where p@k/c@k means pass@k/cons@k.

| Dataset | REINFORCE++ | | | | DPO (Qwen3-1.7B) | | | | | | | |
| | Random | | HAMMER | | Random | | | | HAMMER | | | |
| | 1.7B | 4B | 1.7B | 4B | p@1 | p@10 | p@32 | c@32 | p@1 | p@10 | p@32 | c@32 |
|---|---|---|---|---|---|---|---|---|---|---|---|---|
| AIME24 | 62.1 | 73.1 | 63.7 | 75.7 | 0.7 | 3.0 | 7.3 | 0.0 | 1.0 | 2.7 | 7.3 | 0.0 |
| AIME25 | 38.9 | 50.2 | 43.5 | 55.5 | 1.7 | 5.0 | 6.7 | 0.0 | 2.3 | 8.3 | 12.7 | 0.0 |
| AMC23 | 84.0 | 88.3 | 85.0 | 88.3 | 15.9 | 25.5 | 31.7 | 13.3 | 16.7 | 27.8 | 31.4 | 14.5 |
| Olympiad | 64.3 | 71.9 | 65.3 | 71.5 | 13.8 | 23.7 | 26.7 | 13.1 | 14.5 | 24.9 | 28.5 | 14.1 |

Table 7: Validation training stability for the Qwen3-1.7B model (AMC 2023).

| Steps | Random | | | | HAMMER | | | |
| | Run1 | Run2 | Run3 | Mean $\pm$ Std | Run1 | Run2 | Run3 | Mean $\pm$ Std |
|---|---|---|---|---|---|---|---|---|
| 0 | 49.7 | 48.2 | 49.0 | $49.0 \pm 0.6$ | 48.2 | 48.2 | 48.2 | $48.2 \pm 0.0$ |
| 50 | 53.7 | 55.0 | 56.6 | $55.1 \pm 1.2$ | 60.9 | 60.6 | 60.8 | $60.8 \pm 0.1$ |
| 100 | 63.4 | 63.2 | 60.9 | $62.5 \pm 1.1$ | 60.2 | 64.7 | 62.7 | $62.5 \pm 1.8$ |
| 150 | 58.5 | 63.0 | 60.8 | $60.8 \pm 1.8$ | 63.3 | 65.9 | 61.4 | $63.5 \pm 1.8$ |
| 200 | 57.3 | 57.4 | 57.3 | $57.3 \pm 0.0$ | 59.8 | 63.4 | 61.8 | $61.7 \pm 1.5$ |

curriculum-based reinforcement learning to enhance reasoning and generalization in large language models (Bae et al., 2025; Zeng et al., 2025). Some methods assign difficulty levels to Chain-of-Thought (CoT) annotations (Qiu et al., 2025; Parashar et al., 2025), filter out overly simple or challenging examples, or ensure a balanced distribution of task difficulties. Others employ manually designed curricula that progress from easy to hard tasks after fixed training intervals (Xie et al., 2025; Team et al., 2025). Recently, ADARFT (Shi et al., 2025) dynamically samples based on the relative distance between current and target difficulty levels, while SEC (Chen et al., 2025a) selects samples according to rewards from different difficulty categories. E2H-G and E2H-C (Parashar et al., 2025) use Gaussian/cosine schedulers for arrangement. However, these methods depend on the base model and require evaluation of the model's *pass@k* performance.

**Coreset Selection**  Coreset selection (CS) accelerates training by selecting a compact, representative subset of samples (Koh & Liang, 2017; Schioppa et al., 2021; Sener & Savarese, 2017; Sorscher et al., 2023; Feldman & Langberg, 2016; Lewis & Catlett, 1994; Zhang et al., 2024a). While effective for pruning redundancy, CS inherently faces performance bottlenecks (Mehra et al., 2025). In contrast, our Hamiltonian ordering adopts a curriculum learning view: it leverages semantic diversity to structure training, exposing the model to varied samples early on to promote more curious and stable learning. Unlike CS's focus on reduction, our approach complements it by emphasizing diversity-driven guidance.

**Data Diversity**  The evaluation of text diversity can be categorized into three approaches. (1) $N$-gram based methods (Mishra et al., 2020) utilize lexical statistics through metrics like distinct-$n$ (Song et al., 2024a), self-BLEU (Shu et al., 2019), and ROUGE-L (Wang et al., 2023; Padmakumar & He, 2024) to efficiently quantify surface-level variation. (2) Reference-based methods (Heusel et al., 2017) such as MAUVE (Pillutla et al., 2021) quantify diversity by measuring the distributional divergence between generated texts and a high-quality reference dataset; (3) Transformation-based methods (Miranda et al., 2025) employ learned representations (e.g., from language models) to capture multi-faceted diversity (semantic, syntactic, and stylistic) and summarize it via techniques like clustering (Du & Black, 2019) or eigenvalue computation for VendiScore (Friedman & Dieng, 2023) and its extensions RKE (Jalali et al., 2023) and FKEA (Ospanov et al., 2024), which offer superior flexibility and comprehensiveness but suffer from higher computational complexity.

# E  DATASET AND MODEL DETAILS

Table 8: Summary of datasets with URLs.

| Dataset | Size | URL |
|---|---|---|
| AIME 2024 | 30 | https://huggingface.co/datasets/HuggingFaceH4/aime_2024 |
| AIME 2025 | 30 | https://huggingface.co/datasets/opencompass/AIME2025 |
| AMC 2023 | 83 | https://huggingface.co/datasets/math-ai/amc23 |
| OlympiadBench | 675 | https://huggingface.co/datasets/math-ai/olympiadbench |
| DeepScaleR | 40315 | https://huggingface.co/datasets/agentica-org/DeepScaleR-Preview-Dataset |

Table 9: Details of models used in our experiments.

| Model | Params | URL |
|---|---|---|
| *Reasoning Models* | | |
| Qwen3-1.7B | 1.7B | https://huggingface.co/Qwen/Qwen3-1.7B |
| Qwen3-4B | 4B | https://huggingface.co/Qwen/Qwen3-4B |
| Qwen3-8B | 8B | https://huggingface.co/Qwen/Qwen3-8B |
| Deepseek-R1-Distill-Llama3-8B | 8B | https://huggingface.co/deepseek-ai/DeepSeek-R1-Distill-Llama-8B |
| Qwen3-1.7B-Base | 1.7B | https://huggingface.co/Qwen/Qwen3-1.7B-Base |
| Qwen2.5-Math-1.5B | 1.5B | https://huggingface.co/Qwen/Qwen2.5-Math-1.5B |
| Qwen2.5-1.5B-Instruct | 1.5B | https://huggingface.co/Qwen/Qwen2.5-1.5B-Instruct |
| *Embedding Models* | | |
| all-MiniLM-L6-v2 | 22.7M | https://huggingface.co/sentence-transformers/all-MiniLM-L6-v2 |
| Qwen3-Embedding-4B | 4B | https://huggingface.co/Qwen/Qwen3-Embedding-4B |

