# OpenReview forum: "HAMMER: Hamiltonian Curiosity Augmented Large Language Model Reinforcement"
_ICLR.cc/2026/Conference — Submitted to ICLR 2026_

### Official Review · Reviewer_HLW8 · 2025-10-22

**Soundness:** 2
**Presentation:** 2
**Contribution:** 3
**Rating:** 4
**Confidence:** 4

**Summary:**

The paper introduces a novel curriculum learning schema for reinforcement learning in LLMs. Instead of traditional difficulty-based curricula, HAMMER proposes ordering the training data to maximize diversity.  The authors claim that by presenting the model with a sequence of semantically diverse samples in training, the model is encouraged to explore more of the data space, leading to more stable training, faster convergence, and better accuracies on mathematical reasoning benchmarks. The paper also provides theoretical arguments suggesting this approach tightens generalization bounds while preserving the optimal policy.

**Strengths:**

1. The core idea of framing the data sequencing problem for curriculum learning as a path-finding problem on a semantic similarity graph is novel and creative. Applying a concept from graph theory like the Hamiltonian path problem to RL-based LLM fine-tuning is interesting.
2. The proposed method presents a potentially practical advantage. Difficulty-based curriculum learning often requires expensive data annotation or evaluation runs with powerful proprietary models. HAMMER offers a data-centric alternative that is computationally cheaper, requiring only a one-time forward pass to generate embeddings and a subsequent heuristic search.

**Weaknesses:**

1. Serious incorrectness in the Example 3. In the sole example for Hamiltonian Curiosity Order, the given path $(x_2, x_3, x_5, x_4, x_1)$ has similarity $w_2=-0.8-0.3+0.4+0.7=0.0$, instead of the proposed -0.5. Actually, the targeted paths are  $(x_1, x_3, x_2, x_5, x_4)$ and its reverse $(x_4, x_5, x_2, x_3, x_1)$.
2. Theorems 1 and 2 are general learning theory results that argue for the effectiveness of training on a diverse subset of data. While they correctly suggest that starting with a diverse set of samples is beneficial, they do not provide a compelling reason for why the entire dataset must be structured into a single Hamiltonian sequence. The theory supports diverse mini-batches, but not necessarily this specific and restrictive global ordering.
3. The η-GHS algorithm's performance depends on the "expand factor η". The paper dismisses its importance by stating, "varying η has little impact on overall performance, so we fix η=3", but provides no ablation study or evidence to support this claim. This is a critical omission for a key algorithmic component.
4. The entire method relies on meaningful semantic embeddings from the backbone LLM itself. The paper does not discuss how the quality of these embeddings at the start of training affects the curriculum. If the initial model has poor representations, it could generate a nonsensical or even detrimental training order. An analysis of the robustness of the method to the initial model's capabilities is missing.

Typos:
1. Line 422: As shown in Figure 7 show that... ->Figure 7 shows that
2. Figure 4 (c): varing -> varying

**Questions:**

1. How was η=3 chosen? Please give more detailed explanations and empirical validations. Is the performance robust to this choice?
2. Have you considered or compared HAMMER to a simpler baseline that enforces diversity at the mini-batch level (e.g., using maximal marginal relevance or similar techniques for batch selection) instead of ordering the entire dataset?
3. How sensitive is the HAMMER curriculum to the initial state of the backbone model? Did you experiment with generating the embeddings from a different, external model (e.g., a dedicated sentence-transformer) versus the backbone LLM? Could a curriculum generated by a poorly pre-trained model be detrimental, and does the curriculum need to be recomputed as the model trains and its representations change?

---

> ### Author Response · Authors · 2025-11-19
>
> We sincerely thank you for carefully reading our manuscript and providing constructive feedback. As a reminder, we have highlighted the key revisions in red font and posted a Public Official Comment (titled: Overall Feedback and Summary) for your locating to these modifications.
>
> # W1: Typos
> We inadvertently failed to update the path of $\mathcal{P}_2$ in Example 3 (ambiguous w.r.t. to Figure 1). The optimal path shown in Figure 1 (i.e., $\mathcal{P}^*=\mathcal{P}_2=x_1 \rightarrow x_3 \rightarrow x_2 \rightarrow x_5 \rightarrow x_4$) is indeed correct; its total weight $w_2 = −0.4 − 0.8 + 0.3 + 0.4 = −0.5$ is accurately computed. The outdated path $x_2 \rightarrow x_3 \rightarrow x_5 \rightarrow x_4 \rightarrow x_1$ that remained in the text has a total weight of $−0.8 − 0.3 + 0.4 + 0.7 = 0.0\geq −0.5$, and therefore not optimal. All other descriptions and weight calculations are correct. We have corrected the mentioned typos in the revised version.
>
> # W2/Q2: Suggestion for Diverse Mini-Batch Training
> The idea you mentioned about diverse mini-batch selection is a very interesting, which shares similarities with many dynamic sampling strategies and our HAMMER. Based on your idea, we implemented a version called the Minimum Constraint Batch (MCB) training. Specifically, we set a diversity constraint in each mini-batch training, selecting top-50% of the samples for *backward* through the Farthest Point Sampling strategy (coreset selection in Appendix D). This is essentially a method of choosing the "most diverse samples". **Figure 7(c) in revised manuscript** shows that this diversity-based selection strategy outperforms full-batch training with 50% samples, supporting the conclusions of Theorem 1 and Theorem 2, which means your insight is absolutely correct. However, as mentioned in Section 1, such core-set-style selection may create performance bottlenecks; therefore, HAMMER's global ranking mechanism can bring greater gains than the mini diversity-constrained batch method (MCB). Last but not least, the $\eta$ parameter introduces randomness and adjusts the trade-off between diversity ranking and random ranking, ensuring that HAMMER is not limited to a single global ordering.
>
> # W3/Q1: Random Factor Ablation Study
> $\eta$ is a random factor that governs the selection of the next sample; larger $\eta$ values yield behavior closer to pure random sampling. We conducted an ablation study across different $\eta$ settings. As illustrated in **Figure 6 of the revised manuscript**, $\eta=3$ emerges as the most stable choice, consistently outperforming $\eta=1$ and $\eta=5$ (with no significant divergence observed in the first 200 steps). Notably, $\eta=5$ suffers a performance drop between steps 200–300, consistent with its stronger bias toward random sampling. Conversely, $\eta=1$ exhibits higher variance, presumably because the tighter constraint imposed by the smaller $\eta$ restricts stability.
>
> # W4/Q3: Embedding Model Ablation Study
> We compared the performance of HAMMER's built-in embedding with external encoders BERT and Qwen3-Embedding-4B. HAMMER's built-in embedding consistently outperforms external encoders: BERT and Qwen3-Embedding-4B remain comparable for the first 200 steps but then become unstable, aligning with our statement in Section 2. The embedded approach eliminates the discrepancy between external embeddings and the model with zero additional overhead. Refreshing embeddings and reranking each epoch suppresses deviations; however, in practice, the model often converges within a single epoch. Please refer to **Section 5.5 and Figure 7(a) in the revised manuscript** for details.
>
> On the other hand, poorly trained backbone models/external embedding models can indeed negatively impact HAMMER training. However, in our observations, poorly trained models are often unsuitable for RLVR algorithms as they typically exhibit significant fluctuations and are prone to collapse. Therefore, we recommend performing RLVR training only after a cold start. In our experiments, we provided training and testing for Deepseek-R1-Distill-Llama3-8B, precisely because direct reinforcement learning on Llama3-8B with DAPO/GRPO would lead to collapse, while the former is more stable.

---

> > ### Comment · Reviewer_HLW8 · 2025-11-20
> >
> > The typos in the sole example indeed confused me for a while. But thank you for your detailed review. I will increase my scores.

---

> > > ### Author Response · Authors · 2025-11-21
> > >
> > > We appreciate your time and valuable opinions in your busy schedule. If you have any other concerns, we will be happy to communicate with you further.

---

### Official Review · Reviewer_WNod · 2025-10-26

**Soundness:** 2
**Presentation:** 2
**Contribution:** 2
**Rating:** 4
**Confidence:** 3

**Summary:**

The paper proposes Hammer, an automated curriculum learning method by ordering the training samples via minimum-semantic Hamiltonian path. a training samples are ordered via a minimum-semantic Hamiltonian path. This is intended to induce curiosity, encourage exploration across diverse semantic regions, and avoid overfitting to local clusters. Extensive experiments are provided to show performance gains compared with GRPO, DAPO, and "easy-to-hard" curriculum baselines.

**Strengths:**

- The paper is well-written and easy-to-follow.
- Introducing automated curriculum learning into reinforcement learning for large language models is timely and relevant. The idea of structuring training data through semantic ordering is interesting from an application perspective.
- The paper provides theoretical justifications for main claims.
- The paper evaluates HAMMER with extensive experiments and compares against several baselines, showing consistent improvements for pass@k metrics across multiple Math benchmarks.

**Weaknesses:**

1. The paper claims that "training with the selected Hamiltonian cycle promotes convergence by tightening the generalization bound with a small set of diverse samples”. However, the presented generalization bound (Theorem 1–2) is a standard VC-type inequality that depends only on sample size $n$, confidence $\delta$, and hypothesis class complexity $d$. It's not clear how it is related to sample ordering and diversity of the subset. More justifications will be helpful.
2. Theorem 3 states that maximizing the diversity score $\max_S \mu_{DCS}(S)$ is equivalent to minimizing total pairwise similarity in subset $S$, which is a global diversity objective over a subset $S$. However, the Hamiltonian curiosity order (Definition 3) solves a permutation problem over all samples by minimizing path cost, which minimizes the sum of adjacent similarities, not global pairwise similarity. There seems some fundamental mismatch.
3. The empirical improvements over E2H and other baselines are marginal, and no confidence intervals or error bars are provided. It is therefore unclear if the gains are statistically significant.

**Questions:**

1. Is each subset $S$ selected once or repeatedly per epoch? How many RL update steps are done per subset? Does ordering reset every epoch? Or is it fixed once?
2. Could you empirically compare the diversity metric of samples selected by Hammer, random ordering and other baselines?

---

> ### Author Response · Authors · 2025-11-19
>
> Thank you for your time and valuable feedback. As a reminder, we have highlighted the key revisions in red font and posted a Public Official Comment (titled: Overall Feedback and Summary) for your locating to these modifications.
>
> # W1/W2: Theoretical Analysis
> We discuss the details of theorecal analysis and how it guides the design of HAMMER in **Appendix B.2 of the revised manuscript** (with illustrative figures). We kindly invite you to review it at your convenience to help clarify any misunderstandings. Simply put,
> - **W1**: Theorems 1 and 2 provide learning-theoretic justification for the claim in [1] that "less data can still enable effective reinforcement learning". Rather than verifying whether a given subset satisfies the data-tolerance condition $\gamma\geq 2\rho$, Theorem 1 offers the flexibility to identify a subset $\mathcal{S}$ whose exclusive training does not compromise optimality. Based on this, Theorem 2 shows how training on a diverse subset reduces the generalization error. Together they lay the theoretical basic for HAMMER’s diverse-subset selection.
> - **W2**: Theorem 3 establishes an conversion between diversity metrics and pairwise semantic similarity. Leveraging Theorem 3, HAMMER's ordering is implemented by iteratively/greedly selecting minimal pairwise semantic similarity sample selection. While the global diversity of full dataset remains constant, by redistributing local diversity, prioritizing diverse samples early and finer-grained ones later (see Figure 8(b)), HAMMER translates the static/constant data diversity into a dynamic curriculum, yielding a training paradigm that first emphasizes broad generalization and then tackles specialized challenges.
>
> # W3: Improvement and statistical significance
> In Experimental Setup (Appendix A), we mentioned that during our training process, each validation data point was obtained by averaging the results of 8 samplings (recommended by `verl` [2]). After applying moving average smoothing, HAMMER outperformed all baselines on average. The fact that such improvements ($+3\\%$) could be achieved solely by constructing the training order is significant for RLVR. Taking the Difficulty-based Training ablation (v.s. E2H) as an example, **Figure 5(b) in the revised manuscript** reports the mean pass@8 together with the standard-deviation scatter plot, attesting to the stability of the validation process and the statistical significance of the gains.
>
> # Q1: Concerns about Subset
> As described in **Appendix B.2 of the revised manuscript**, the subset of HAMMER corresponds to the greedy diverse (minimum pairwise semantic similarity) selected samples. Regarding the concerns:
> - Each of our subsets is used only once in a training epoch.
> - All subsets are updated only once.
> - We refresh embeddings and reorder samples after every epoch; yet one epoch usually suffices for convergence in practice.
>
> # Q2: Comparison with Other CL Methods
> In the main experiment (**Table 1 of the revised manuscript**), we demonstrate the superiority of HAMMER over random ordering. Furthermore, we have expanded 2 comparisons with the latest curriculum-learning baselines, including ADARFT [3], SEC [4], and E2H-G/C [5]. Some of the comparative results are shown in the table below, more results/details can be found in **Section 5.5 and Table 5 of the revised manuscript**.
>
> **Comparison with ADARFT: Qwen2.5-Math-1.5B**
> |Data Split|Alg|GSM8K|AIME 2024|AMC 2023|OlympiadBench|Avg.|Source|
> |-|-|-|-|-|-|-|-|
> |full|Zero-shot|43.5|5.3|22.8|18.2|22.458|Ours|
> |skew-difficult|ADARFT|74.0|12.1|55.0|20.4|40.4|[1]|
> |uniform|ADARFT|74.5|12.1|57.5|22.0|41.5|[1]|
> |skew-easy|ADARFT|74.5|9.2|55.0|19.9|39.7|[1]|
> |full|HAMMER|77.8|15.3|58.2|23.0|43.6|Ours|
>
> **Comparison with SEC: Qwen2.5-1.5B-Instruct (GSM8K)**
> |Alg|Trivial|Easy|Med|Hard|Overall|Source|
> |-|-|-|-|-|-|-|
> |Zero-shot|-|-|-|-|73.2|[2]|
> |SEC|98.1|95.3|87.0|50.3|77.8|[3]|
> |E2H-G|97.6|94.7|89.0|51.8|78.7|[3]|
> |E2H-C|98.0|95.3|83.9|46.6|75.7|[3]|
> |HAMMER|-|-|-|-|78.2|Ours|
>
> Results show that, HAMMER outperforms ADARFT, E2H-C and is slightly below E2H-G. However, HAMMER avoids the need for difficulty annotations, offering a more low-cost/efficient training strategy (Table 4).
>
> # References
> [1] Xuefeng Li, Haoyang Zou, and Pengfei Liu. Limr: Less is more for rl scaling, 2025a. URL https://arxiv.org/abs/2502.11886.
>
> [2] Sheng et al., Hybridflow: A flexible and efficient rlhf framework. arXiv preprint arXiv: 2409.19256, 2024.
>
> [3] Taiwei Shi, Yiyang Wu, Linxin Song, Tianyi Zhou, and Jieyu Zhao. Efficient reinforcement finetuning via adaptive curriculum learning, 2025.
>
> [4] Xiaoyin Chen, Jiarui Lu, Minsu Kim, Dinghuai Zhang, Jian Tang, Alexandre Piché, Nicolas Gontier, oshua Bengio, and Ehsan Kamalloo. Self-evolving curriculum for llm reasoning, 2025a. URL https://arxiv.org/abs/2505.14970.
>
> [5] Parashar et al., Curriculum reinforcement learning from easy to hard tasks improves LLM reasoning. URL https://doi.org/10.48550/arXiv.2506.06632.

---

> > ### Comment · Reviewer_WNod · 2025-11-22
> >
> > Thank you for the detailed reply. I have a few follow-up questions:
> > 1. For the statistical significance, I am not asking for the validation process. I am asking the error bars across independent training runs. But I understand this can be costly to run during the rebuttal period.
> > 2. I am still confused about the claims: "Based on this, Theorem 2 shows how training on a diverse subset reduces the generalization error." Also, similar statement in line 299-300. The theorem 1 and 2 supports that subset selection is reasonable, i.e., training on a subset satisfying the data-tolerance condition can preserve the optimal policy, and enjoy the generalization bound. But It's not clear why training on a diverse subset reduce the generalization bound or accelerate convergence.
> > 3. The proposed training strategy permutes subsets and uses each subset only once, rather than optimizing the policy on a fixed selected subset. The theorems seem to support subset selection but not the claim that the particular permutation schedule improves performance relative to other shuffling strategies. In line 1012, you refer to “Figure 4(a)” as evidence. Can you clarify what Figure 4(a) shows and how it supports this claim?

---

> ### Author Response · Authors · 2025-11-23
>
> We sincerely thank you for your early response and all the effort you have devoted!
>
> # Q1: Training Error Bar for Static Significance
> We understand your concerns regarding error bars and recognize their importance in ensuring the reproducibility of the work. We also appreciate your understanding of the challenges we face during the Rebuttal stage. Indeed, it is difficult for us to conduct multiple training runs for each experimental setup. However, at this stage, we have made every effort to provide three training runs for the Qwen3-1.7B model (validate on AMC 2023).
> |Type|Step|Run1|Run2|Run3|Mean|±Std|
> |-|-|-|-|-|-|-|
> |Random|0|49.7|48.2|49.0|49.0|±0.6|
> ||50|53.7|55.0|56.6|55.1|±1.2|
> ||100|63.4|63.2|60.9|62.5|±1.1|
> ||150|58.5|63.0|60.8|60.8|±1.8|
> ||200|57.3|57.4|57.3|57.3|±0.0|
> |HAMMER|0|48.2|48.2|48.2|48.2|±0.0|
> ||50|60.9|60.6|60.8|60.8|±0.1|
> ||100|60.2|64.7|62.7|62.5|±1.8|
> ||150|63.3|65.9|61.4|63.5|±1.8|
> ||200|59.8|63.4|61.8|61.7|±1.5|
>
> The results show that despite the deviations between the three training sessions, the improvement of HAMMER remains relatively significant. For a visual comparison of the three training runs with error bars, please refer to our anonymous repo (link: line 543); the corresponding error-bar plot is provided as `error_bars.png`.
>
> # Q2: Why Diverse Subset Can Reduce Generalization Bound?
> We appreciate your understanding of the statement regarding "training on subsets that meet data tolerance conditions can preserve the optimal strategy and achieve generalization bounds." Regarding your concerns about "**why training on diverse subsets can reduce generalization bounds**," we can understand them from the following two points:
> - **Intuitive Understanding**: As mentioned in [1], training with diverse datasets can enhance model generalization. Moreover, it is an wide observation that data diversity fosters robustness in deep learning. An example: Suppose we conduct two training, the first using similar samples and the second using diverse samples. Clearly, the latter will exhibit superior generalization capabilities (generalization bounds) because the model is exposed to a wider variety of information.
> - **Theoretical Combination**: The generalization bound in Theorem is given by
>     $$
>     \Delta_\pi \leq \mathcal{O} \left(\sqrt{\frac{d\log(n/d)+\log(1/\delta)}{n}}\right),
>     $$
>     which is inversely proportional to $d$. Simply put, when the dataset size $n$ is fixed, using a more diverse subset increases the effective complexity $d$ because the data cover a broader infomation of the data space, which directly leads to a smaller generalization upper bound.
>
> # Q3：From Diverse Subset Selection to Diverse Subset Ordering
>
> **(1) From Diverse Subset Selection to Diverse Subset Ordering**
>
> As you mentioned, the theorems illustrate how diverse subset selection contributes to the reduction of generalization bounds. However, it's worth noting: **training on the entire dataset can be viewed as multiple diverse subset selections (discussed in lines 1018-1022)**. Based on this, we might divide the training processes into three categories:
>
> ```shell
> # Generalization Bound
> ^
> |A B C       C
> | A   B              C
> |  A      B               C
> |    A         B             C
> |      A            B          C
> |         A              B        C
> |            A               B      C
> |                A              B    C
> |                       A       A  B  C
> +--------------------------------------->
> # (Diverse Subset Selection) Training Steps
> A: HAMMER (diverse to similar)
> B: Random Shuffle
> C: HAMMER reversed (similar to diverse)
> ```
> According to the Q2 explanation, by prioritizing diverse subsets early in training, HAMMER accelerates the reduction of the generalization bound in the earlier stage. In practice, this "generalize broadly before specializing" strategy enhances RLVR training for LLMs, promoting faster stabilization, which is particularly important given the high variance and instability typically observed in the early stages of RLVR [2].
>
> **(2) A typo: Line 1012 references "Figure 4(a)"**
>
> Thanks for the reminder. We mistakenly wrote "Figure 4(a)" in line 1012, but please refer to Figure 5(a) instead. In this experiment, we investigated the three aforementioned training procedures (random, diverse-first then similar, similar-first then diverse). We discussed the comparison of these three approaches in lines 425-429. As discussed, an earlier and faster decline in the generalization bound helps the RLVR process achieve stable improvement while minimizing instability caused by "local optima."
>
> # References
> [1] Yuchang Zhu, Huizhe Zhang, Bingzhe Wu, Jintang Li, Zibin Zheng, Peilin Zhao, Liang Chen, and Yatao Bian. Measuring diversity in synthetic datasets. CoRR, abs/2502.08512, 2025.
>
> [2] Zhipeng Chen, Xiaobo Qin, Youbin Wu, Yue Ling, Qinghao Ye, Wayne Xin Zhao, and Guang Shi. Pass@k training for adaptively balancing exploration and exploitation of large reasoning models, 2025c.

---

> > ### Comment · Reviewer_WNod · 2025-11-24
> >
> > Thanks for the quick response. Most of my concerns have been addressed. I’m prepared to raise my score, and hope that the authors will incorporate the additional discussions and results in their revision.

---

> > > ### Author Response · Authors · 2025-11-24
> > >
> > > We appreciate your active participation and patient discussion. The discussion process with you is smooth and very meaningful. We will further improve the theoretical explanation and relevant experimental demonstration based on the discussion with you. If you have any other concerns, we will be happy to discuss with you! Good luck!

---

### Official Review · Reviewer_uEBG · 2025-10-31

**Soundness:** 4
**Presentation:** 3
**Contribution:** 3
**Rating:** 6
**Confidence:** 3

**Summary:**

The paper proposes HAMMER, a curriculum learning schema for reinforcement learning of LLMs, which structures training by ordering samples based on minimum semantic similarity (formulated as a Hamiltonian path) to promote early exposure to diverse examples. Theoretical analysis demonstrates that this approach preserves the optimal policy and tightens generalization bounds, while experiments on mathematical reasoning benchmarks show consistent 3–4% accuracy gains and improved sample efficiency over baselines using difficulty-based or random sample orderings.​

**Strengths:**

- Novelty: Transferring diversity metrics from static dataset analysis for RL curriculum to address the exploration-exploitation imbalance common in difficulty-based curricula.
- Theoretical section justifies generalization bounds and shows that sample diversity ordering does not compromise learning optimality, even as it reduces variances.
- Performance: Improving upon the baseline using randomly shuffled samples.
- Efficiency: Training with faster and better convergence compared to baselines.

**Weaknesses:**

- The complexity of Algorithm 1 is $O(n^2)$, thus, the scalability of Hamiltonian ordering as the number of training samples grows is not fully explored, especially when models and datasets scale up beyond several thousand samples.
- Include embedding model ablations comparing internal vs. external representations could be helpful to support the use of backbone LLM's embeddings (L162-166).
- While the performance gained is good given the model sizes of 1.7B and 4B, I wonder if we can see this benefit in a larger model.
- Minor: add citations for training and evaluation datasets.

**Questions:**

- Is the performance improvement mainly attributable to the use of the backbone LLM’s own embeddings, or could similar gains be realized using pre-trained external sentence encoders?
- While using the backbone LLM's embeddings might align well semantically, would it introduce bias or restrict exploration to the model’s current latent space structure?
- Could HAMMER be applied to other RL methods (e.g., DPO)?
- Have authors ever considered mixing HAMMER (semantic diversity ordering) and easy-to-hard progression?

---

> ### Author Response · Authors · 2025-11-19
>
> Thank you for your feedback and for your deep understanding of our work. As a reminder, we have highlighted the key revisions in red font and posted a Public Official Comment (titled: Overall Feedback and Summary) for your locating to these modifications.
>
> # W1: Algorithm Scalability
> Our training dataset comprises over 40,000 samples, a substantial scale in the RLVR field. We measured the time required for each step of generating sample sequences using 1 CPU and 4 NVIDIA A100 GPUs. The results demonstrate that, across models of varying scales, our method significantly outperforms difficulty-based ordering approaches in runtime when handling 40,000 samples. A concise comparison is provided below (Table 4 in the revised manuscript gives detailed runtime for each part):
>
> **Time comparison for training dataset ordering (minutes)**
> ||Qwen3-1.7B|Qwen3-4B|Qwen3-8B|Deepseek-R1-Distill-Llama3-8B|
> |-|-|-|-|-|
> |HAMMER|124.48|136.33|152.02|155.88|
> |E2H|195.78|334.48|538.51|542.32|
>
> Furthermore, when the sample size $n\gg40000$, we recommend using clustering to reduce the time overhead of the similarity matrix and ordering parts. We have provided this functionality in the supplementary materials/anonymous code (i.e., enabling the ``--kmeans`` option in ``sample_similarity_matrix.py``).
>
> # W2/Q1/Q2: Embedding Model Ablation Ablation Study
> Please refer **Section 5.5 and Figure 7(a) in revised manuscript** for the embedding model ablation study. HAMMER’s backbone embedding consistently outperforms external encoders: BERT and Qwen3-Embedding-4B match HAMMER in the first 200 steps but eventually becomes unstable, which aligns with the statement in Section 2. Additionally, we believe that using backbone embedding can better eliminate the discrepancy between external embeddings and the trained model, without introducing additional costs. Results show that HAMMER’s stable gains stem from early-stage sample diversity computed with internal embeddings, which fosters rather than restricts exploration in RL. After each training epoch, we recompute the embeddings and reorder the samples using the latest model to mitigate the bias you mentioned; however, in practice, the model often converges within a single epoch.
>
> # W3: Model Scaling
> **Table 1 in the revised manuscript** now includes additional results for both the isogenic Qwen3-8B and the heterogenic Deepseek-R1-Distill-Llama3-8B models, with their GRPO training dynamics illustrated in Figure 4. The outcomes confirm that HAMMER continues to accelerate RL learning as model scale increases.
>
> # W4: Citation
> In the revised manuscript, we have added comprehensive citations for all datasets and models used, with detailed information and links provided in the Appendix E.
>
> # Q3: Other RL Algorithms
> Our extensive experiments (Table 1) primarily demonstrate the role of HAMMER in RLVR. We think DPO also can enhance learning efficiency through diversity, yet DPO is not RLVR and requires preference data during training. How to represent <query, preferences> data is a very interesting topic. However, due to time limits, we plan to include experiments with methods like DPO in future. Nevertheless, we additionally provide the pass@10 evaluation of the REINFORCE++ [1] algorithm trained on DeepScaleR.
>
> ||REINFORCE++(Random)||REINFORCE++(HAMMER)||
> |-|-|-|-|-|
> ||Qwen3-1.7B|Qwen3-4B|Qwen3-1.7B|Qwen3-4B|
> |AIME 2024|62.1|73.1|63.7|75.7|
> |AIME 2025|38.9|50.2|43.5|55.5|
> |AMC 2023|84.0|88.3|85.0|88.3|
> |OlympadBench|64.3|71.9|65.3|71.5|
>
> We will consider more RL algorithms in the future and conduct experiments on larger models.
>
> # Q4: Conbination E2H with HAMMER
> The combination of E2H and HAMMER is an interesting idea. To examine the effect of combining E2H and HAMMER, we construct a new curriculum order (E2H&HAMMER) for DeepScaleR using difficulty as the primary key and HAMMER’s diversity order as the secondary key, and validate it on AIME 2024. **Figure 7(b)  in the revised manuscript** shows that this combined ordering underperforms both individual methods and resembles worse/random.
>
> # References
> [1] Hu, J., Liu, J.K., Xu, H., & Shen, W. (2025). REINFORCE++: Stabilizing Critic-Free Policy Optimization with Global Advantage Normalization.

---

> ### Author Response · Authors · 2025-11-21
> **Comparison on DPO**
>
> To showcase HAMMER’s impact on non-RLVR algorithms such as DPO [2], we fine-tuned Qwen3-1.7B with DPO on the Infinity-Preference [1] dataset while keeping every HAMMER hyper-parameter identical to the main experiment. Infinity-Preference supplies preference triplets *<query, (chosen, rejected)>*; we curriculum-sort them by *query* embedding for HAMMER. The empirical findings are presented below:
>
> **Comparison on Qwen3-1.7B with DPO Algorithm**
> |Dataset|total|DPO|||||HAMMER|||||
> |-|-|-|-|-|-|-|-|-|-|-|-|
> |||pass@1|pass@10|pass@32|cons@32|Avg.|pass@1|pass@10|pass@32|cons@32|Avg.|
> | AIME24 | 30 | 0.7 | 3.0 | 7.3 | 0.0 | 2.8 | 1.0 | 2.7 | 7.3 | 0.0 | 2.8 |
> | AIME25 | 30 | 1.7 | 5.0 | 6.7 | 0.0 | 3.3 | 2.3 | 8.3 | 12.7 | 0.0 | 5.8 |
> | AMC | 83 | 15.9 | 25.5 | 31.7 | 13.3 | 21.6 | 16.7 | 27.8 | 31.4 | 14.5 | 22.6 |
> | Olympiad | 673 | 13.8 | 23.7 | 26.7 | 13.1 | 19.3 | 14.5 | 24.9 | 28.5 | 14.1 | 20.5 |
>
> **Absolute Delta**
> | Dataset | Δpass@1 | Δpass@10 | Δpass@32 | Δcons@32 | ΔAvg |
> |-|-|-|-|-|-|
> | AIME24 | +0.3 | −0.3 | +0.0 | +0.0 | +0.0 |
> | AIME25 | +0.7 | +3.3 | +6.0 | +0.0 | +2.5 |
> | AMC | +0.8 | +2.3 | −0.2 | +1.2 | +1.0 |
> | Olympiad | +0.7 | +1.2 | +1.8 | +1.0 | +1.2 |
>
> From an average perspective, HAMMER's performance on DPO remains consistent with optimization effects of RLVR algorithms like GRPO/DAPO, indicating that HAMMER maintains stable effectiveness on DPO.
>
> # References
> [1] https://huggingface.co/datasets/BAAI/Infinity-Preference
>
> [2] Rafailov, R., Sharma, A., Mitchell, E., Manning, C. D., Ermon, S., & Finn, C. (2023). Direct preference optimization: Your language model is secretly a reward model. Advances in neural information processing systems, 36, 53728-53741.

---

> ### Author Response · Authors · 2025-11-27
>
> Dear Reviewer,
>
> First of all, thank you for your insightful suggestions.  As the rebuttal phase is drawing to a close, considering that any experiments in the field may take some time to complete, we would like to confirm whether our responses have addressed your concerns.  Your support would be of great help in improving the quality of our work.
>
> Have a good day!
>
> Your Authors

---

### Official Review · Reviewer_h9kM · 2025-11-02

**Soundness:** 2
**Presentation:** 3
**Contribution:** 2
**Rating:** 4
**Confidence:** 3

**Summary:**

This paper introduces HAMMER, a new curriculum learning schema for reinforcement learning with LLMs. Instead of ordering training samples by difficulty, it orders them by semantic diversity. It constructs a Hamiltonian Curiosity Order, a data sequence with minimum semantic similarity, so that the model encounters the most diverse samples first. It also proposes a heuristic algorithm to compute the Hamiltonian Curiosity Order. Theoretical analysis shows that the proposed method preserves the optimal policy and tightens generalization bounds. Experiments on several benchmarks also demonstrates its effectiveness.

**Strengths:**

1. The proposed HAMMER method can achieve the data ordering using only semantic embeddings, without requiring extra annotations or difficulty metrics, making it more practical and scalable.
2. It provides theoretical analysis that supports the effectiveness of the proposed method.
3. The experimental results show that HAMMER outperforms the baseline methods on several benchmarks.

**Weaknesses:**

1. Most of the experiments compare HAMMER only with a baseline that uses random data ordering. There is only one experiment comparing HAMMER with E2H and H2E. It lacks comprehensive comparisons with other curriculum learning methods that use different data ordering strategies.
2. The theoretical analysis does not provide insight into why the Hamiltonian Curiosity Order is better than other possible ordering strategies, e.g., random sampling.
3. The novelty of the proposed method is somewhat limited, as it mainly combines existing ideas of curriculum learning and semantic embeddings without introducing significant new techniques.

**Questions:**

1. Can you provide more experiments comparing HAMMER with other curriculum learning methods to better demonstrate its effectiveness?
2. Can you provide additional theoretical analysis or discussion on why ordering by the proposed Hamiltonian Curiosity Order leads to better performance compared to other ordering strategies, or under what conditions can it work better?

---

> ### Author Response · Authors · 2025-11-19
>
> We sincerely appreciate your careful review of our manuscript. Your feedback has been invaluable in improving our research. As a reminder, we have highlighted the key revisions in red font and posted a Public Official Comment (titled: Overall Feedback and Summary) for your locating to these modifications.
>
> # W1/Q1: Comparison with Other CL Methods
> In the main experiment (**Table 1 of the revised manuscript**), we demonstrate the superiority of HAMMER over random ordering. Furthermore, we have expanded 2 comparisons with the latest curriculum-learning baselines. To ensure fairness given differing model and dataset settings, in our latest manuscript, we conduct 2 comparisons with the most recent related methods, including ADARFT [1], SEC [5], E2H-G/C [3]. Some of the comparative results are shown in the table below, more results/details can be found in **Section 5.5 and Table 5 of the revised manuscript**.
>
> **Comparison with ADARFT: Qwen2.5-Math-1.5B**
> |Data Split|Alg|GSM8K|AIME 2024|AMC 2023|OlympiadBench|Avg.|Source|
> |-|-|-|-|-|-|-|-|
> |full|Zero-shot|43.5|5.3|22.8|18.2|22.458|Ours|
> |skew-difficult|ADARFT|74.0|12.1|55.0|20.4|40.4|[1]|
> |uniform|ADARFT|74.5|12.1|57.5|22.0|41.5|[1]|
> |skew-easy|ADARFT|74.5|9.2|55.0|19.9|39.7|[1]|
> |full|HAMMER|77.8|15.3|58.2|23.0|43.6|Ours|
>
> **Comparison with SEC: Qwen2.5-1.5B-Instruct (GSM8K)**
> |Alg|Trivial|Easy|Med|Hard|Overall|Source|
> |-|-|-|-|-|-|-|
> |Zero-shot|-|-|-|-|73.2|[2]|
> |SEC|98.1|95.3|87.0|50.3|77.8|[3]|
> |E2H-G|97.6|94.7|89.0|51.8|78.7|[3]|
> |E2H-C|98.0|95.3|83.9|46.6|75.7|[3]|
> |HAMMER|-|-|-|-|78.2|Ours|
>
> HAMMER outperforms ADARFT in first experiment. In the second experiment, HAMMER outperforms E2H-C and is slightly below E2H-G. However, HAMMER avoids the need for difficulty annotations, offering a low-cost/efficient training strategy (Table 4).
>
> # W2/Q2: Theoretical Analysis
> We have discussed three theorems in detail in Appendix B.2 (with example). Theorems 1/2, combined with [4] that demonstrate that training with diverse subsets can achieve good performance compared to random sampling. Theorem 3 builds upon Theorems 1/2 by providing the conditions for converting diversity ranking to semantic similarity ranking. HAMMER is a dynamic local diversity curriculum reordering based on diversity, so its effectiveness depends on the overall high diversity of the data. We selected three subsets (5000 samples) of the DeepScaleR  dataset with different diversity levels, trained Qwen3-1.7B using HAMMER GRPO, and the evaluation results on the AIME 2024 dataset are as follows:
> |$\mu_{\text{DCS}}$|1.25|1.30|1.35|
> |-|-|-|-|
> |Acc|32.0|35.7|36.5|
>
> The accuracy positively correlated with the overall dataset diversity metric $\mu_{\text{DCS}}$, which indicates that global dataset diversity has a positive impact on HAMMER's ordering by local diversity.
>
> # W3: Novelity
> Recently, curriculum reinforcement learning approaches for large language models have generally been based on difficulty settings, often requiring pre-defined difficulty annotations [1-3]. To the best of our knowledge, HAMMER is the first method to propose leveraging diversity for curriculum reinforcement learning. Building on the effectiveness of local diversity in training, we transform static diversity metrics into dynamic curriculum sequences and propose a low-cost yet efficient ordering algorithm. Experiments and theories jointly demonstrate that diversity-based ranking is not only concise and elegant but also comprehensively outperforms traditional difficulty-dependent methods in effectiveness.
>
> # References
> [1] Taiwei Shi, Yiyang Wu, Linxin Song, Tianyi Zhou, and Jieyu Zhao. Efficient reinforcement finetuning via adaptive curriculum learning, 2025.
>
> [2] Qwen2.5 technical report. URL https://arxiv.org/abs/2412.15115.
>
> [3] Parashar et al., Curriculum reinforcement learning
> from easy to hard tasks improves LLM reasoning. URL https://doi.org/10.48550/arXiv.2506.06632.
>
> [4] Xuefeng Li, Haoyang Zou, and Pengfei Liu. Limr: Less is more for rl scaling, 2025a. URL
> https://arxiv.org/abs/2502.11886.
>
> [5] Xiaoyin Chen, Jiarui Lu, Minsu Kim, Dinghuai Zhang, Jian Tang, Alexandre Piché, Nicolas Gontier, oshua Bengio, and Ehsan Kamalloo. Self-evolving curriculum for llm reasoning, 2025a. URL https://arxiv.org/abs/2505.14970.

---

> > ### Author Response · Authors · 2025-11-28
> > **Looking for your early reply :)**
> >
> > Dear Reviewer,
> >
> > We sincerely thank you for your insightful and professional feedback. We have carefully revised the manuscript and would be grateful if you could confirm whether the revisions address your concerns.
> >
> > Sincerely,
> > The Authors

---

### Author Response · Authors · 2025-11-19
**Overall Feedback and Summary**

We sincerely appreciate the reviewers for their careful review and feedback, which has provided valuable suggestions for our research. Our gratitude also extends to the PC/AC/SAC and ICLR 2026 for offering this opportunity for open communication.

# Revision Summary
We have submitted the revised manuscript, with major changes highlighted in **RED**. Below is an index of the key revisions and corresponding locations in the LATEST VERSION of the manuscript.

| Item | Content | Index | Source |
|-|-|-|-|
| Typos of Example 3 | We forgot to update the path of $\mathcal{P}$ in Example 3 (causing ambiguity with Figure 1); the remaining description and weight calculations are correct. | Example 3 (line 199) | HLW8 |
| Curriculum-order Construction Time | We provide HAMMER and a difficulty-based method for curriculum-order construction. Results show HAMMER is significantly faster. | Appendix C, Table 4 (line 1062) | uEBG |
| Main Results | Added isomorphic Qwen3-8B and heterogeneous Deepseek-R1-Distill-Llama3-8B results on top of Qwen3-1.7B/4B. | Table 1 (line 324-355) | uEBG |
| Training Dynamics | Provided training curves for Qwen3-8B, Deepseek-R1-Distill-Llama3-8B and Qwen3-1.7B. | Figure 4 (line 409-417) | — |
| Zero-shot Results | Aligned with main experiments, we report zero-shot results of the newly added models. | Table 2 (line 474-485) | — |
| Ablation on $\eta$ | Supplemented the missing $\eta=3$ ablation experiment. | Figure 6 & Discussion (line 459-466) | HLW8 |
| Comparison with Other CL Methods | Due to differing models/data across works, we conduct two fair comparison groups on Qwen2.5-Math-1.5B / Qwen2.5-1.5B-Instruct against ADARFT [1], SEC [3], E2H-G/C [2]. | Discussion (line 492-499); Table 5; Appendix E | h9kM, WNod |
| Embedding Model Ablation | Incorporated external BERT / Qwen3-Embedding-4B as sentence encoders into HAMMER. | Line 500-504, Figure 7(a) (line 518-525) | uEBG, HLW8 |
| Additional Exploratory Experiments | (1) Combine difficulty-based curriculum learning with HAMMER; (2) Dynamic diversity sampling. | (1) Discussion (line 506-509), Figure 7(b); (2) Discussion (line 511-517), Figure 7(c) | uEBG, HLW8 |
| Theoretical Discussion | Detailed theoretical analysis of HAMMER, including three theorems and their connections to the method (Discussion 1-3). | Appendix B.2 | h9kM, WNod, HLW8 |

We appreciate and welcome all feedback you provide. If you have any other concerns, please feel free to contact us.

# References
[1] Taiwei Shi, Yiyang Wu, Linxin Song, Tianyi Zhou, and Jieyu Zhao. Efficient reinforcement finetuning via adaptive curriculum learning, 2025.

[2] Shubham Parashar, Shurui Gui, Xiner Li, Hongyi Ling, Sushil Vemuri, Blake Olson, Eric Li, Yu Zhang, James Caverlee, Dileep Kalathil, and Shuiwang Ji. Curriculum reinforcement learning from easy to hard tasks improves LLM reasoning. CoRR, abs/2506.06632, 2025. doi: 10.48550/ARXIV.2506.06632. URL https://doi.org/10.48550/arXiv.2506.06632.

[3] Xiaoyin Chen, Jiarui Lu, Minsu Kim, Dinghuai Zhang, Jian Tang, Alexandre Piché, Nicolas Gontier, oshua Bengio, and Ehsan Kamalloo. Self-evolving curriculum for llm reasoning, 2025a. URL https://arxiv.org/abs/2505.14970.

---

### Author Response · Authors · 2025-11-29
**Sumary of Before Rebuttal and Our Work**

Dear PC/AC/SAC/Reviewers,

We sincerely appreciate the time and effort each reviewer has dedicated, as well as their constructive feedback. The discussions with you have been both an important foundation for improving our work and a great honor for us. Despite the twists and turns caused by the "information leak," the ICLR committee did not let us down. We are grateful for their timely and insightful decisions, which ensured the rebuttal process continued smoothly. This has been a fascinating and unique experience in our academic journey. Most importantly, we must express our highest respect to the PC, SAC, and ACs (both the original and the new ones), as this incident ultimately placed greater pressure on your workload. Yet, it is through your efforts that ICLR 2026 has become something truly special.

To alleviate the review burden on the ACs and to better present our (thorough/innovative) work, we have summarized the following points:

# Summary about Our *HAMMER*

Reinforcement learning with LLMs often exhibits high variance and unstable convergence, particularly in the early stages of training [4]. Previous work [1-3] has introduced curriculum reinforcement learning to mitigate these issues. However, these approaches primarily rely on methods like filtering, sorting, or dynamic sampling based on difficulty-level annotations, which typically suffer from the following drawbacks:
- The difficulty level of many tasks is hard to define/evaluate (hard is a relative concept for different models).
- Difficulty-based curriculum learning requires costly resources for difficulty annotation or inference evaluation (e.g., 40,000+ training samples for pass@32 testing, which takes nearly a week).
- The difficulty distribution is often noncontinuous/uneven ([1] and Figure 9(c), not a continuous transition from easy to hard), making difficulty-based curriculum learning less effective.

In our work, we *firstly* propose *Hamiltonian Curiosity Augmented Large Language Model Reinforcement (HAMMER)*, which innovatively leverages diversity (a common static metric in pretraining), by modifying local diversity during training to create a dynamically training process. Diversity (transformed from similarity), is easier to compute, and has a smoother distribution. Our theory establishes the transformation between diversity and similarity, further demonstrating that early-stage diversity training in RLVR leads to faster generalization and improved stability. Extensive experiments validate the effectiveness of HAMMER.

# Summary about Before Rebuttal

In the initial rebuttal, reviewers raised some important suggestions:

|Reviewer|Main Comments Summary|Details Link|
|-|-|-|
|h9kM|Insufficient experimental comparison (only random ranking + few baselines); theoretical analysis is unclear, making it difficult to elucidate HAMMER’s principles and role.|[*check*](https://openreview.net/forum?id=OfIPPocM4H&noteId=wPAm5wA7Qx)|
|uEBG|Need to expand experiment/model scale; clarify the root cause of performance gains (source of embeddings); explore method potential (other algorithms or combination strategies).|[*check*](https://openreview.net/forum?id=OfIPPocM4H&noteId=HD3XtuzsMP)|
|WNod|Theory is incomplete; experiments lack confidence intervals, raising doubts about reproducibility and statistical significance|[*check*](https://openreview.net/forum?id=OfIPPocM4H&noteId=fd4ibLv5Ae)|
|HLW8|Theoretical flaws; insufficient parameter ablation; recommend ablation experiments on how data representations affect the method|[*check*](https://openreview.net/forum?id=OfIPPocM4H&noteId=M5R49y31kQ)|

We acknowledge all the reviewers' suggestions, mainly focusing on more theoretical explanations and richer experiments. The discussion was productive, with the concerns of WNod/HLW8 largely resolved, leading to an increase in scores (4->6), occurring well before any information leakage (21 Nov 2025, 06:36 and 25 Nov 2025, 06:13). Due to time constraints, the discussions with h9kM/uEBG have not yet concluded, but there are many overlaps (theoretical and experimental concerns) discussed with WNod/HLW8. We hope we have addressed the reviewers' concerns.

We have summarized the major modifications we made earlier at [*link*](https://openreview.net/forum?id=OfIPPocM4H&noteId=K9SiqlOAXz) for your reference and marked the changes in RED in our latest manuscript. Detailed score changes are as follows:

|Score|h9kM|uEBG|WNod|HLW8|Avg.|
|-|-|-|-|-|-|
**Before Rebuttal**
|Soundness|2|4|2|2|2.5|
|Presentation|3|3|2|2|2.5|
|Contribution|2|3|2|3|2.5|
|Rating|4|6|4|4|4.5|
|Confidence|3|3|3|4|3.25|
**After Rebuttal**
|**Rating**|4|6|6|6|**5.5**|

PS: Soundness/Presentation/Contribution/Confidence remain the same as before.

Finally, we would like to once again express gratitude to all reviewers and PC/AC/SAC. Regardless of the final result, we respect and understand the considerations and challenges involved.

Wishing you all the best!

---

> ### Author Response · Authors · 2025-11-29
> **References**
>
> # References
>
> [1] Taiwei Shi, Yiyang Wu, Linxin Song, Tianyi Zhou, and Jieyu Zhao. Efficient reinforcement finetuning via adaptive curriculum learning, 2025.
>
> [2] Shubham Parashar, Shurui Gui, Xiner Li, Hongyi Ling, Sushil Vemuri, Blake Olson, Eric Li, Yu Zhang, James Caverlee, Dileep Kalathil, and Shuiwang Ji. Curriculum reinforcement learning from easy to hard tasks improves LLM reasoning. CoRR, abs/2506.06632, 2025. doi: 10.48550/ARXIV.2506.06632. URL https://doi.org/10.48550/arXiv.2506.06632.
>
> [3] Xiaoyin Chen, Jiarui Lu, Minsu Kim, Dinghuai Zhang, Jian Tang, Alexandre Piché, Nicolas Gontier, oshua Bengio, and Ehsan Kamalloo. Self-evolving curriculum for llm reasoning, 2025a. URL https://arxiv.org/abs/2505.14970.
>
> [4] Zhipeng Chen, Xiaobo Qin, Youbin Wu, Yue Ling, Qinghao Ye, Wayne Xin Zhao, and Guang Shi. Pass@k training for adaptively balancing exploration and exploitation of large reasoning models, 2025c. URL https://arxiv.org/abs/2508.10751.

---

### Meta-Review · Area_Chair_KKro · 2026-01-06

**Summary:**

Summary of the main concerns:

1. Mismatch between theory and the proposed approach (h9kM, WNod, HLW8)
2. Comparison with more curriculum methods missing (h9kM, WNod)
3. Missing comparisons with other embedding methods. (uEBG, HLW8)
4. Concern about scalability of the approach to larger datasets. (uEBG)
5. Concern about scalability of the approach to larger models. (uEBG)
6. Concern about the statistical significance of the results. (WNod)

**Reviewer Concerns:**

To my understanding, most of the important concerns seem largely resolved. From the partial discussion, the reviewers seem convinced by the authors' additional explanation of the connection between the theory and the approach. For concern (2), the new results comparing against other baselines seem convincing, although HAMMER does not outperform all the new baselines. I believe the new results provided address concerns 3-6 well.

**Reviewer Scores:**

Two of the reviewers were willing to raise the scores before the discussion ended. I expect all the scores would have ended up in the range of 6 for this paper. It is worth noting that none of the reviewers was particularly championing this paper, and the presentation, soundness and contribution scores were mostly "fair". The general sense seems to be that this is a theoretically heavy paper for the contribution it makes.

---

### Decision · Program_Chairs · 2026-01-26

Reject